# FIMP: Foundation Model-Informed Message Passing for Graph Neural Networks

## Abstract

Foundation models have achieved remarkable success across many domains, relying on pretraining over vast amounts of data. Graph-structured data often lacks the same scale as unstructured data, making the development of graph foundation models challenging. In this work, we propose Foundation-Informed Message Passing (FIMP), a Graph Neural Network (GNN) message-passing framework that repurposes existing pretrained non-textual foundation models for graph-based tasks. We show that the self-attention layers of foundation models can effectively be leveraged on graphs to perform cross-node attention-based message-passing. Our model is evaluated across diverse domains on image networks, single-cell RNA sequencing, and fMRI brain activity recordings in finetuned and zero-shot settings. FIMP outperforms strong baselines, demonstrating that it can effectively leverage state-of-the-art foundation models in graph tasks.

## 1 Introduction

Foundation models have emerged as a new paradigm in artificial intelligence, shifting from narrow, task-specific training to large-scale pretraining of more generalized models (Brown et al., 2020). Through pretraining on vast amounts of data, foundation models serve as a base model which can be adapted to a variety of downstream tasks (Bommasani et al., 2021). Pretraining is typically done in self-supervised fashion through autoregressive language modeling (Radford et al., 2018), masked language/image modeling (Devlin et al., 2018; Chen et al., 2020), or other self-supervised objectives. Standard foundation models have emerged in fields such as Natural Language Processing (NLP) with BERT (Devlin et al., 2018), GPT-3 (Brown et al., 2020), and CLIP (Radford et al., 2021), as well as in Computer Vision (CV) (Yuan et al., 2021). More recently, fields such as single-cell RNA sequencing and neuroscience have also seen the emergence of large-scale foundation models such as scGPT (Cui et al., 2023), Geneformer (Theodoris et al., 2023), and BrainLM (Ortega Caro et al., 2023), representing a new frontier in foundation model research.

Despite the success of foundation models in domains such as language and vision, training and leveraging such models for graph-structured data remains a significant challenge. One key difficulty is the relative scarcity of large-scale, publicly available graph-structured data compared to unstructured data, which limits the capacity to pretrain foundation models specifically for graph tasks. In single-cell RNA sequencing (scRNAseq) data, for instance, advances in sequencing technology have fueled an exponential growth in available unstructured single-cell transcriptomes (Svensson et al., 2018), however spatial sequencing methods which preserve the spatial organization of cells within the tissue during sequencing lag behind in scale and resolution. Furthermore, traditional Graph Neural Networks (GNNs) tokenize nodes as single embedding vectors, whereas transformer-based foundation models represent inputs as sequences of feature tokens, processing them at a more granular level. Prominent examples include gene tokenization in scGPT (Cui et al., 2023) and image patching in Vision Transformers (ViTs) (Dosovitskiy et al., 2020; He et al., 2022). This feature-level tokenization separates traditional GNNs from foundation models and remains underutilized in graph-based settings. **Bridging the gap between traditional GNNs and pretrained foundation models, and by extension unstructured and structured data, remains an open challenge.**

Existing works have increasingly explored how pretrained foundation models, particularly Large Language Models (LLMs), can be applied to graph-based tasks, primarily in the context of text-attributed graphs. One-for-All (Liu et al., 2023) used LLMs to encode text-attributed graphs for a

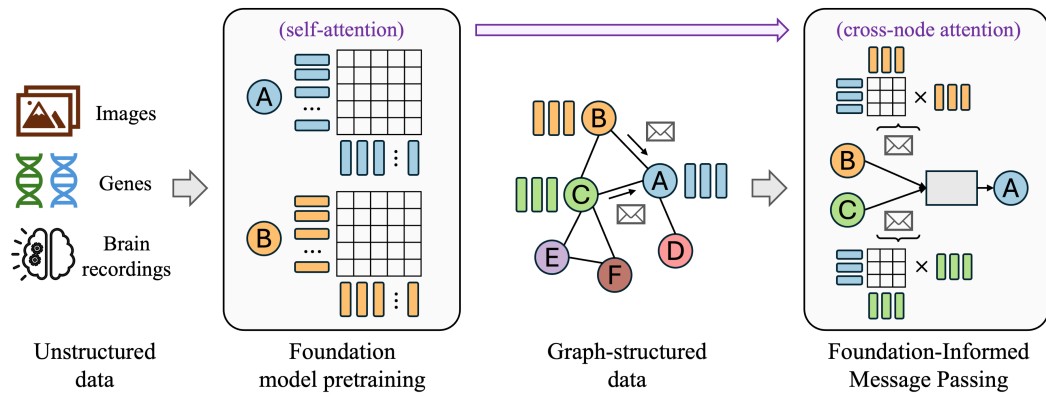

Figure 1: The proposed framework for FIMP. Pre-existing foundation models, pretrained on vast amounts of unstructured data, are repurposed into message creation modules by adapting their self-attention layers for cross-node attention between node feature sequences.

GNN model, enabling the GNN to do node, edge, and graph-level classification tasks jointly. Talk Like a Graph (Fatemi et al., 2023), NLGraph (Wang et al., 2023), and GPT4Graph (Guo et al., 2023) evaluated LLM reasoning capabilities on graph reasoning benchmarks. These approaches have made significant strides in applying LLMs to text-attributed graphs. However, **non-textual foundation models remain largely underexplored in non-textual graph settings**, leaving significant opportunities for leveraging models like scGPT and BrainLM in graph-based tasks.

To address these challenges, we propose Foundation-Informed Message Passing (FIMP), a novel message-passing framework that repurposes existing pretrained non-textual foundation models for message-passing on graphs. FIMP unifies node tokenization between GNNs and foundation models by viewing nodes as sequences of feature tokens, and introduces a cross-node attention-based message creation module which can be learned from scratch or initialized from pretrained foundation models. We evaluate FIMP across several domains, including street-view image classification (Antequera et al., 2020), spatial transcriptomics, and fMRI brain activity recordings, incorporating state-of-the-art (SOTA) models like ViTs for images (Dosovitskiy et al., 2020), scGPT for scRNAseq (Cui et al., 2023), and BrainLM for brain recordings (Ortega Caro et al., 2023). FIMP demonstrates improvements over strong GNN baselines, highlighting the potential of repurposing non-textual foundation models for graph-based tasks. Additionally, FIMP demonstrates zero-shot embedding capabilities on image networks by leveraging pretrained ViTs (Dosovitskiy et al., 2020), achieving competitive performance without additional training.

**Contributions.** In summary, our work makes the following key contributions:

1. We introduce FIMP, a message-passing framework that leverages pretrained non-textual foundation models for graph-based tasks.

2. We evaluate FIMP across diverse domains including images, spatial transcriptomics, and fMRI recordings, repurposing SOTA non-textual foundation models as message creators.

3. We demonstrate FIMP's zero-shot embedding capabilities using pretrained ViTs on image networks, showing that non-textual foundation models can effectively handle graph-based tasks without task-specific training.

## 2 PRELIMINARIES

### 2.1 GRAPH NEURAL NETWORKS

Graph Neural Networks are a versatile class of models designed to operate over graph-structured data. The core idea of GNNs is to learn node and/or edge attributes through iterative local aggrega-

tion steps, which is commonly implemented through Message-Passing Neural Networks (MPNNs) (Gilmer et al., 2017). Below we define our notations for describing GNNs.

Let $G = (V, E)$ denote a graph with a set of nodes $V$ and edges $E$. Each node has an input feature vector $\vec{x}_i \in \mathbb{R}^f$, where $f$ is the number of input features per node. GNNs iteratively pass messages between neighboring nodes, and in the process use both node features and graph structure to learn node representations $\vec{h}_i \in \mathbb{R}^d$, where $d$ is the hidden dimension of node embeddings. After $K$ message-passing iterations, node representation $\vec{h}_i$ will contain information from its $K$-hop neighborhood within the graph. The general update rule for the $k$-th layer can be represented as:

$$\vec{h}_{\mathcal{N}(i)}^{(k)} = \text{AGGREGATE}^{(k)}\big(\{\vec{h}_j^{(k-1)}, \forall j \in \mathcal{N}(i)\}\big) \tag{1}$$

$$\vec{h}_i^{(k)} = \text{COMBINE}^{(k)}\big(\vec{h}_i^{(k-1)}, \vec{h}_{\mathcal{N}(i)}^{(k)}\big), \tag{2}$$

where $\mathcal{N}(i)$ denotes the neighborhood of node $i$ and $h_i^{(k)}$ is the representation of node $i$ in layer $k$. The choice of AGGREGATE and COMBINE vary among different GNN architectures, with the constraint that AGGREGATE should be a permutation-invariant aggregator. A readout function is used to map learned node representations into predictions for feature, node, or graph-level tasks.

## 2.2 ATTENTION-BASED GNNS

Attention-based GNNs, such as Graph Attention Networks (GATs) (Veličković et al., 2017), improve the standard aggregation mechanism by learning the attention coefficients between nodes. In these models, the AGGREGATE function from equation 1 is replaced by an attention mechanism, which computes weighted combinations of neighboring node embeddings based on learned attention scores

$$e_{ji} = a(\mathbf{W}\vec{h}_i || \mathbf{W}\vec{h}_j) \tag{3}$$

$$\alpha_{ji} = \text{softmax}_j(e_{ji}) \tag{4}$$

where $\alpha_{ji}$ represents the final normalized attention coefficient between nodes $i$ and $j$, $a$ is a learned attention mechanism shared across all node pairs, and $\mathbf{W}$ represents a shared weight matrix.

However, it is important to note that **FIMP fundamentally differs from attention-based GNNs like GATs and graph transformers** (covered in detail in section D). In these models, each node is represented by a single embedding $\vec{h}_i$, and attention is computed at the node level, producing scalar attention values between pairs of neighboring nodes. In FIMP, nodes are represented as sequences of feature vectors (tokens) rather than single embeddings, more aligned to pretrained transformers. FIMP's message-passing is driven by cross-node attention between these token sequences, allowing for richer, more granular interactions between neighboring nodes. This token-based message creation process is unique to FIMP and is described in further detail in section 3.

## 2.3 FOUNDATION MODELS

Foundation models are generalized Deep Learning models which have been pretrained on large amounts of data, and which can be finetuned for a variety of downstream tasks. In this work, we focus on non-textual foundation models, which define a tokenization procedure for continuous-valued data and typically do pretraining using a masked reconstruction objective. In single-cell RNA sequencing, for instance, scGPT (Cui et al., 2023) tokenizes an input cell as a sequence of gene tokens, and learns a gene embedding table analogous to word embeddings learned in LLMs. Pretraining is done through a masked gene expression prediction objective. In the image domain, ViT-based architectures (Dosovitskiy et al., 2020; He et al., 2022) encode images as a sequence of patches, and similarly for fMRI brain activity recordings, BrainLM (Ortega Caro et al., 2023) tokenizes segments of brain activity signal per brain region into tokens.

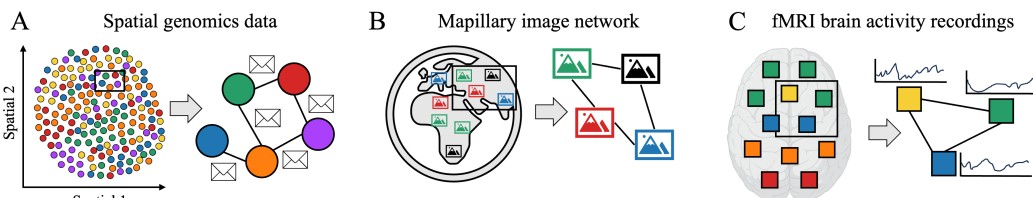

Figure 2: Graph structure present in real-world datasets. (A) In spatially resolved RNA transcriptomics, cells are connected to adjacent cells in the 2D tissue section. (B) In the Mapillary street-view image dataset (Antequera et al., 2020), images form a geographical proximity graph. (C) For fMRI recordings, the brain is parcellated into 424 regions, which are connected using a K-nearest neighbors graph based on the 3D spatial coordinates of each brain region.

## 3 FOUNDATION-INFORMED MESSAGE PASSING

We propose a novel message-passing framework, depicted in Figure 1, that uses pretrained non-textual foundation models to generate messages between neighboring nodes in a graph. This leverages the pretrained knowledge of the foundation model to inform message-passing, allowing for pretraining on unstructured data before training on less-abundant graph-structured data.

### 3.1 NODE TOKENIZATION

To align the node tokenization procedure in FIMP with the tokenization used by pretrained transformers, we introduce a transformation function, $\tau$. This function ensures that a given node's features are tokenized into a sequence of feature vectors, similar to how transformers tokenize input entities into input sequences. Specifically, $\tau$ takes as input node features $X_i \in \mathbb{R}^{f \times c}$, where $f$ is the number of features per node and $c$ is the dimensionality of each featuree. It outputs a sequence of $f$ $d$-dimensional feature vectors representing node $i$. By aligning the tokenization in FIMP with the tokenization scheme of pretrained foundation models, we reduce distribution shift in token representation when applying these models to graph-structured data. A general formulation of $\tau$ is:

$$H_i = \tau(X_i) = \texttt{COMBINE}(X_i \mathbf{W}, P) \in \mathbb{R}^{f \times d} \tag{5}$$

where $\mathbf{W}$ is a $c \times d$ learned projection into a $d$-dimensional feature vector, $P \in \mathbb{R}^{f \times d}$ are positional encodings for each feature, and $\texttt{COMBINE}$ represents element-wise addition. The dataset-specific instantiations of node tokenization for scRNAseq and fMRI brain recordings are further detailed in Appendix section B.

### 3.2 MESSAGE CREATION

Our objective is to formulate message creation between two nodes such that pretrained foundation models can be leveraged to create the messages while fitting into the rest of the message-passing framework. Our key observation is that transformer-based foundation models operate using self-attention over sequences of feature tokens (depicted in Figure 1), and contain learned attention weights per layer which are trained to highlight important interactions between feature tokens. Message creation between neighboring node feature sequences can be viewed as a problem of highlighting relevant information which source node $j$ must pass to destination node $i$, and thus the pretrained attention weights can be repurposed for message creation between two nodes.

We define a cross-node attention-based message creation module which takes as input node feature sequences $H_i$ and $H_j$, and outputs a message token sequence $H_{ji}$ which will be passed from node $j$ to node $i$:

$$Q = H_i \mathbf{W}_Q, \; K = H_j \mathbf{W}_K, \; V = H_j \mathbf{W}_V, \tag{6}$$

$$H_{ji} = \mathtt{softmax}\left(\frac{QK^\top}{\sqrt{d}}\right)V \qquad (7)$$

where $\mathbf{W}_Q$, $\mathbf{W}_K$, and $\mathbf{W}_V$ are learned weight matrices which parameterize the attention mechanism. Note that the attention weights can be randomly initialized and learned from scratch, or initialized from pretrained attention weights. Messages $H_{ji}$ can then be aggregated and used to complete the regular message passing aggregation and update steps, with each node represented by a sequence of feature tokens rather than a single embedding vector. The full algorithm is detailed in Algorithm 1.

We note that the cross-attention-based message passing operation in FIMP is fundamentally different from other attention-based GNNs. FIMP is the first method that uses feature-based cross-node attention to construct messages for message passing on graphs. In contrast, attention-based GNNs, particularly GATs and Graph Transformers, do node-level attention and learn scalar attention coefficients between nodes. An overview of attention-based GNNs is provided in the Related Works (section D), along with a summary of key differences with FIMP.

---

**Algorithm 1** FIMP

---

**Require:** Graph $G = (V, E)$, input features $X_i \in \mathbb{R}^{f \times c}$
  $H_i^0 \leftarrow \tau(X_i)$
  **for** $k = 1...K$ **do**
    **for** node $i \in V$ **do**
      **for** node $j \in \mathcal{N}(i)$ **do**
        $Q = H_i^{(k-1)}\mathbf{W}_Q$
        $K = H_j^{(k-1)}\mathbf{W}_K$
        $V = H_j^{(k-1)}\mathbf{W}_V$
        $H_{ji}^{(k)} = \mathtt{softmax}\left(\frac{QK^\top}{\sqrt{d}}\right)V$
      **end for**
      $H_{\mathcal{N}(i)}^{(k)} = \underset{j \in \mathcal{N}(i)}{\mathtt{AGGREGATE}}\left(H_{ji}^{(k)}\right)$
      $H_i^{(k)} = \mathtt{COMBINE}(H_i^{(k-1)}, H_{\mathcal{N}(i)}^{(k)})\mathbf{W}$
    **end for**
  **end for**

---

### 3.3 LEVERAGING NON-TEXTUAL FOUNDATION MODELS

In its base formulation, cross-attention message passing can be done with a simple cross-attention mechanism which is learned from scratch during training. We denote this base version of our architecture as FIMP-base in our experiments. Pretrained foundation models, however, can be repurposed to do the message creation in order to leverage their pretraining over vast amounts of unstructured data. Given a pretrained foundation model $\mathcal{F}$ with learned attention weights per each transformer layer, we adapt the self-attention mechanism in each layer to do cross attention between node feature sequences from neighboring nodes. This adaptation is done in each layer by using the pretrained $\mathbf{W}_Q$, $\mathbf{W}_K$, and $\mathbf{W}_V$ weights to project both the source and destination node feature sequences $H_j$ and $H_i$, and computing the scaled dot product attention outlined in equation 7. The final hidden representation output of the foundation model is then taken as the message $H_{ji}$.

## 4 EXPERIMENTS

In this section, we demonstrate the effectiveness of our proposed framework on a diverse range of tasks in graph-structured settings: (i) gene expression reconstruction and cell type classification on spatial transcriptomics datasets, (ii) image classification on the Mapillary street-view image dataset, and (iii) brain activity reconstruction on fMRI brain recordings from the UK Biobank (UKB) dataset (Miller et al., 2016). The graph structure inherent in each of these datasets is depicted in Figure 2. We show that FIMP allows for the effective integration of pretrained non-textual foundation models into a message-passing framework on graphs.

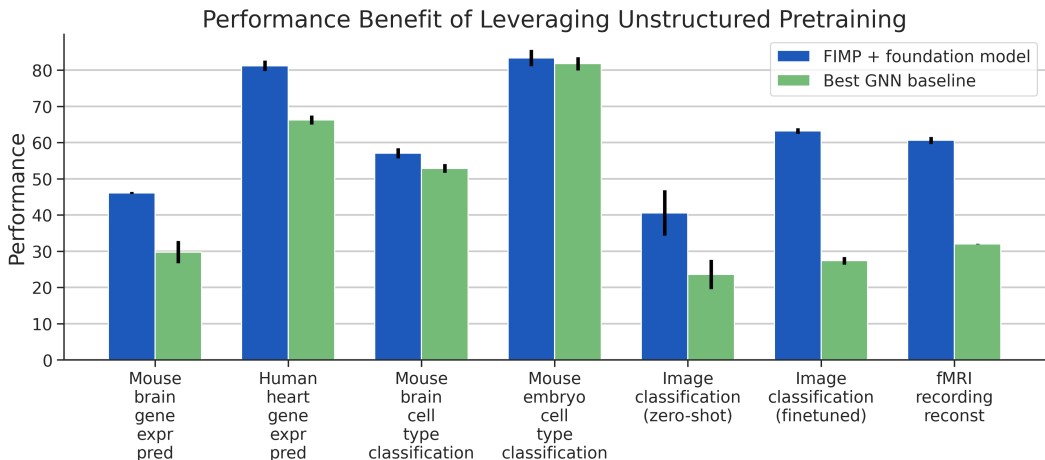

Figure 3: Performance summary across different tasks for FIMP + foundation model versus the best traditional GNN baseline. FIMP improves over traditional GNNs across multiple datasets, highlighting the benefits of leveraging foundation models pretrained on unstructured data.

## 4.1 DATASETS

**Spatial transcriptomics.** We benchmark FIMP on gene expression prediction and cell type classification using three publicly-available spatial transcriptomics datasets. The Slideseq-V2 spatial transcriptomics dataset (Stickels et al., 2021) is a mouse hippocampus dataset consisting of $41,786$ cells and $4,000$ genes, with $14$ different cell types. A second spatial dataset of human heart tissue was obtained from the 10X Genomics public spatial data repository, consisting of $4,247$ cells each with $36,601$ measured genes. A third spatial dataset, SeqFISH (Lohoff et al., 2020), consists of $15,000$ cells and $342$ genes taken from mouse embryo tissue sections. For all spatial transcriptomics datasets, we follow standard preprocessing and normalization procedures for RNA sequencing data, including count normalization and log transformation (Haque et al., 2017). Full dataset details are in Appendix section A.

**Mapillary image dataset.** The Mapillary planet-scale image dataset (Antequera et al., 2020) is a dataset of 750,000 street-view images collected from over 170 countries around the world. Images are 1000-2000 pixels in height and width, originating from a variety of cameras and conditions depicting natural landscapes and buildings. Each image has a recorded latitude and longitude coordinate, forming a geographical proximity graph where each node represents a full image, connected to nearby image nodes if they are within 10 miles of one another. We evaluate FIMP on a task where the aim is to classify the country of origin based on the visual features of each image node and its neighborhood. We train on 100,000 training images, and test on the predefined 10,000 test image set, with country labels determined for each image based on its latitude and longitude coordinates.

**fMRI brain activity recordings.** The UK Biobank dataset (Miller et al., 2016) comprises of 76,296 task-based and resting-state functional MRI (fMRI) recordings from 41,986 patients aged 40 to 69 years old. All recordings went through standard preprocessing steps for fMRI recordings (Salimi-Khorshidi et al., 2014; Abdallah, 2021), and was parcellated into 424 brain regions using the AAL-424 atlas (Nemati et al., 2020). We apply robust scaling per brain region by subtracting the median and dividing by the interquartile range computed across subjects. Our training set comprised of $60,000$ recordings, with the rest reserved for validation and test.

## 4.2 EXPERIMENTAL SETUP

All models were implemented in Pytorch Geometric (Fey & Lenssen, 2019) and Pytorch (Paszke et al., 2019), and trained using the Adam optimizer (Kingma & Ba, 2014). Flash Attention (Dao et al., 2022) is used to improve the computational footprint during message passing. Hyperparameter tuning was done through a grid search over standard values for learning rate, dropout, attention

Table 1: Gene expression prediction results on the mouse hippocampus and human heart spatial transcriptomics datasets. Performance is reported across 5 runs in terms of MSE and $R^2$. FIMP outperforms baseline methods on predicting gene expression on both datasets.

| Method | Mouse Hippocampus | | Human Heart | |
|---|---|---|---|---|
| | MSE ($\downarrow$) | $R^2$($\uparrow$) | MSE ($\downarrow$) | $R^2$($\uparrow$) |
| GCN | 0.0211 ± 0.0018 | 0.0236 ± 0.0457 | 0.0045 ± 0.00019 | 0.3368 ± 0.04453 |
| GraphSAGE | 0.0181 ± 0.0012 | 0.1853 ± 0.0306 | 0.0054 ± 0.00033 | 0.2080 ± 0.01973 |
| GAT | 0.0201 ± 0.0008 | 0.0905 ± 0.0233 | 0.0043 ± 0.00023 | 0.3468 ± 0.02313 |
| GIN | 0.0175 ± 0.0009 | 0.1707 ± 0.0424 | 0.0025 ± 0.00029 | 0.6625 ± 0.01269 |
| GraphMAE | 0.0178 ± 0.0006 | 0.1538 ± 0.0254 | 0.0024 ± 0.00016 | 0.6589 ± 0.01715 |
| GPS | 0.0149 ± 0.0012 | 0.2977 ± 0.0308 | 0.0024 ± 0.00031 | 0.6538 ± 0.01043 |
| scGPT | 0.0169 ± 0.0007 | 0.2087 ± 0.0191 | 0.0209 ± 0.00072 | 0.0229 ± 0.01757 |
| FIMP-base (ours) | *0.0134 ± 0.0009* | *0.3815 ± 0.0226* | *0.0021 ± 0.00003* | *0.6955 ± 0.02048* |
| FIMP + ViT (ours) | 0.0128 ± 0.0010 | 0.3506 ± 0.0452 | 0.0042 ± 0.00089 | 0.4026 ± 0.08102 |
| FIMP + GenePT (ours) | 0.0129 ± 0.0005 | 0.4058 ± 0.0302 | 0.0013 ± 0.00023 | 0.7952 ± 0.01430 |
| FIMP + scGPT (ours) | **0.0119 ± 0.0008** | **0.4612 ± 0.0029** | **0.0011 ± 0.00008** | **0.8119 ± 0.01428** |

Table 2: Cell type classification results on the mouse hippocampus and embryo spatial transcriptomics datasets. Performance is reported in terms of accuracy and F1-score. FIMP outperforms baseline models at predicting cell types.

| Method | Mouse Hippocampus | | Mouse Embryo | |
|---|---|---|---|---|
| | Accuracy ($\uparrow$) | F1-score ($\uparrow$) | Accuracy ($\uparrow$) | F1-score ($\uparrow$) |
| GCN | 47.59 ± 3.788 | 0.445 ± 0.050 | 74.23 ± 1.250 | 0.720 ± 0.008 |
| GraphSAGE | 51.81 ± 3.229 | 0.495 ± 0.036 | 80.77 ± 3.071 | 0.793 ± 0.031 |
| GAT | 46.21 ± 3.110 | 0.442 ± 0.031 | 71.07 ± 1.452 | 0.690 ± 0.014 |
| GIN | 52.71 ± 0.421 | 0.507 ± 0.008 | 75.51 ± 1.398 | 0.743 ± 0.012 |
| GPS | *52.89 ± 1.176* | *0.510 ± 0.008* | *81.77 ± 3.175* | *0.813 ± 0.038* |
| FIMP-base | 49.04 ± 1.215 | 0.464 ± 0.019 | 81.35 ± 2.285 | 0.807 ± 0.026 |
| scGPT | 53.50 ± 0.424 | 0.518 ± 0.005 | 82.93 ± 0.419 | 0.820 ± 0.005 |
| FIMP-scGPT | **57.05 ± 1.393** | **0.554 ± 0.004** | **83.33 ± 2.250** | **0.821 ± 0.022** |

dropout, and weight decay. For all experiments, a 24GB NVIDIA GPU (RTX3090 or A5000) was used for training. Experimental setup details for specific datasets are provided in the Appendix C.

**Foundation models**. For experiments on single-cell datasets, the scGPT (Cui et al., 2023) whole-human checkpoint is incorporated for message creation in FIMP-scGPT, consisting of a 12-layer transformer with 54 million parameters. scGPT is pretrained using a masked gene expression prediction objective on over 33 million cells from a diverse array of human tissues and organs. The pretrained gene embedding table is also utilized from the pretrained scGPT checkpoint, representing pretrained knowledge about gene identities in transcriptomics datasets. Additionally, we also utilize the gene embeddings obtained by GenePT (Chen & Zou, 2023), which are GPT-3.5 embeddings of gene function descriptions based on biomedical literature, as another pretrained gene embedding experiment. For image classification, a standard ViT (Dosovitskiy et al., 2020) with 12 transformer layers and 86 million parameters is used as a message creator. The patch encoder from the ViT is also reused from the ViT embedding module. For experiments on fMRI brain recordings, the BrainLM (Ortega Caro et al., 2023) model was used, which consists of a Masked Autoencoder transformer with an 8-layer encoder and 4-layer decoder, totaling 26 million parameters.

**Baselines**. For both supervised and self-supervised tasks, we compare FIMP against popular message-passing GNN architectures, including GCN (Kipf & Welling, 2016), GraphSAGE (Hamilton et al., 2017), Graph Attention Networks (GATs) (Veličković et al., 2017), and Graph Isomorphism Networks (GINs) (Xu et al., 2018). We also compare FIMP against more recent GNN architectures, namely GraphMAE (Hou et al., 2022), a masked graph autoencoder model, and GPS Graph Transformer (Rampášek et al., 2022), a SOTA graph transformer framework. For supervised

Table 3: Image classification results on the Mapillary street-view image dataset. FIMP significantly improves over baseline models in image classification, and creates zero-shot embeddings of the image network on par with trained GNN baseline models.

| Setting | Method | Accuracy (↑) | F1-score (↑) |
|---------|--------|--------------|--------------|
| Finetuned | GCN | $23.9 \pm 1.152$ | $0.182 \pm 0.0151$ |
| | GraphSAGE | $22.2 \pm 1.703$ | $0.164 \pm 0.0129$ |
| | GAT | $22.9 \pm 0.596$ | $0.189 \pm 0.0042$ |
| | GIN | $26.4 \pm 1.240$ | $0.254 \pm 0.0143$ |
| | GraphMAE | $15.8 \pm 0.828$ | $0.083 \pm 0.0056$ |
| | GPS | $27.4 \pm 1.046$ | $0.268 \pm 0.0157$ |
| | FIMP-base (ours) | $38.6 \pm 1.174$ | $0.422 \pm 0.0170$ |
| | ViT | $56.5 \pm 3.187$ | $0.597 \pm 0.0065$ |
| | FIMP-ViT (ours) | $\mathbf{63.2 \pm 0.764}$ | $\mathbf{0.684 \pm 0.0076}$ |
| Zero-shot | Majority class | $17.0 \pm 3.162$ | – |
| | GraphSAGE | $23.6 \pm 4.037$ | $0.129 \pm 0.0309$ |
| | ViT | $34.0 \pm 3.391$ | $0.282 \pm 0.0389$ |
| | FIMP-ViT (ours) | $\mathbf{40.6 \pm 6.269}$ | $\mathbf{0.371 \pm 0.0550}$ |

classification tasks, we additionally compare to the pretrained foundation model in each domain, which does not take graph structure as input and instead treats each node as an individual sample.

## 4.3 RESULTS

**Spatial transcriptomics**. Table 1 contains results for gene expression prediction on the human heart and mouse hippocampus datasets. From these results, we observe that FIMP-base, trained from scratch with a randomly initialized cross-attention layer as a message creator, is able to outperform baseline GNNs at predicting masked gene expression values. We attribute this to improved gene tokenization, with the learned gene embedding table capturing information about different genes from the data. When we leverage pretrained gene embeddings learned on unstructured data, either from GenePT (Chen & Zou, 2023) or scGPT (Cui et al., 2023) (denoted as FIMP-GenePT and FIMP-scGPT, respectively), we observe further increases in gene expression prediction performance. Interestingly, we note that using an out-of-domain foundation model such as ViT as the message creator does not improve performance, suggesting that performance improvements are not trivially caused by increased model capacity, and rather depend on the pretraining domain being sufficiently aligned with the graph features.

Table 2 contains results for cell type classification on the mouse hippocampus and embryo spatial transcriptomics datasets. We note that in this supervised classification task, FIMP-scGPT achieves the highest classification performance on both datasets.

**Image classification**. Table 3 summarizes results for image classification on the Mapillary image dataset. We observe that FIMP-base outperforms baseline GNNs by over $10\%$ due to its improved tokenization of image patches, despite being learned from scratch. The best performance is obtained by FIMP-ViT, which utilizes a pretrained ViT (Dosovitskiy et al., 2020) for cross-node message creation. A breakdown of training time for each model is provided in Appendix section F.

**Zero-shot node embedding**. We furthermore explore a zero-shot setting for embedding image networks, to evaluate the capability of FIMP to leverage the pretrained ViT model without any graph-specific training. We embed subgraphs of the Mapillary dataset with FIMP, and compare it to embeddings generated by a randomly initialized GraphSAGE model (Hamilton et al., 2017) and the ViT model itself with no graph structure, with 400 image embeddings obtained per model. We evaluate the quality of embeddings by training a linear classifier on $75\%$ of the embeddings and predicting labels for the remaining $25\%$. We observe that FIMP-ViT is able to generate zero-shot embeddings which get over $40\%$ classification accuracy, on par with finetuned baseline GNNs despite having no graph-specific training. This strongly indicates that FIMP is able to effectively leverage pretrained non-textual foundation models, and enables exciting zero-shot application scenarios which were previously not possible with non-textual foundation models operating on unstructured data.

Table 4: Brain activity reconstruction results on the UK Biobank dataset. Performance is reported across 5 runs. FIMP improves upon baselines by 25.8%, with a further improvement of 2.8% by leveraging BrainLM (Ortega Caro et al., 2023) for message creation.

| Method | Masking Strategy | MSE ($\downarrow$) | $R^2$ ($\uparrow$) |
|---|---|---|---|
| GCN | Replace noise | $0.554 \pm 0.00002$ | $0.189 \pm 0.00003$ |
| | Fill in mean | $0.513 \pm 0.00019$ | $0.248 \pm 0.00028$ |
| | Linear interpolation | $0.535 \pm 0.00137$ | $0.217 \pm 0.00200$ |
| GraphSAGE | Replace noise | $0.534 \pm 0.00107$ | $0.218 \pm 0.00157$ |
| | Fill in mean | *$0.464 \pm 0.00039$* | *$0.320 \pm 0.00057$* |
| | Linear interpolation | $0.500 \pm 0.00094$ | $0.268 \pm 0.00138$ |
| GAT | Replace noise | $0.548 \pm 0.00004$ | $0.197 \pm 0.00007$ |
| | Fill in mean | $0.505 \pm 0.00005$ | $0.260 \pm 0.00007$ |
| | Linear interpolation | $0.527 \pm 0.00052$ | $0.229 \pm 0.00076$ |
| GIN | Replace noise | $0.564 \pm 0.00131$ | $0.174 \pm 0.00192$ |
| | Fill in mean | $0.533 \pm 0.00185$ | $0.220 \pm 0.00271$ |
| | Linear interpolation | $0.559 \pm 0.00061$ | $0.181 \pm 0.00090$ |
| GraphMAE | Replace noise | $0.582 \pm 0.00070$ | $0.147 \pm 0.00103$ |
| | Fill in mean | $0.544 \pm 0.00030$ | $0.203 \pm 0.00044$ |
| | Linear interpolation | $0.573 \pm 0.00091$ | $0.160 \pm 0.00134$ |
| GPS Graph Transformer | Replace noise | $0.577 \pm 0.00279$ | $0.154 \pm 0.00408$ |
| | Fill in mean | $0.547 \pm 0.01030$ | $0.198 \pm 0.01506$ |
| | Linear interpolation | $0.557 \pm 0.01034$ | $0.184 \pm 0.01512$ |
| FIMP-base | Tokenization + PE | $0.288 \pm 0.00713$ | $0.578 \pm 0.01043$ |
| FIMP-BrainLM | Tokenization + PE | $\mathbf{0.267 \pm 0.00493}$ | $\mathbf{0.606 \pm 0.00972}$ |

**fMRI recording reconstruction**. Table 4 summarizes results for fMRI recording reconstruction on the UK Biobank (Miller et al., 2016) dataset. FIMP-base improves upon baseline GNNs by 25% in terms of reconstruction performance on masked brain signals, with a further performance improvement of around 3% from leveraging the pretrained BrainLM (Ortega Caro et al., 2023) model for cross-node message creation.

## 4.4 ABLATION STUDIES

To better understand the contributions of the pretrained foundation model embeddings versus the FIMP architecture, we conducted an ablation study on the Mapilary image classification task. Specifically, we compared the performance of GNN baseline models using embeddings from a pretrained ViT model as input, allowing us to separate the effects of the foundation model embeddings from the performance improvements provided by FIMP's message-passing architecture. Table 5 presents the results of the ablation study. While we observed that the foundation model embeddings enhanced the performance of the baseline GNNs, FIMP still consistently outperformed all baselines. This suggests that FIMP's advantage lies not only in its use of foundation models, but also in its ability to repurpose the pretrained models to facilitate effective message-passing across the graph. Importantly, we highlight that non-textual foundation models cannot natively take graph-structured data as input, but within FIMP, these pretrained foundation models can be meaningfully applied in graph-based learning beyond simple embedding-based inputs.

## 5 CONCLUSIONS, LIMITATIONS, AND FUTURE RESEARCH

In this work, we introduce Foundation-Informed Message Passing (FIMP), a message-passing framework which repurposes pretrained non-textual foundation models for message-passing on graphs. Our approach represents the first broad exploration of utilizing non-textual pretrained foundation models graph settings. FIMP demonstrates improved performance over baselines across multiple tasks in image networks, spatial transcriptomics data, and fMRI brain activity recordings, con-

Table 5: Ablation study comparing FIMP with GNN baseline models with foundation model embeddings as input on the Mapillary image classification task. While foundation model embeddings do enhance performance for some GNNs, FIMP-ViT notably outperforms all baselines by effectively utilizing ViT pretrained weights for message-passing.

| Model Input | Method | Accuracy ($\uparrow$) | F1-score ($\uparrow$) |
|---|---|---|---|
| Image Pixels | GCN | $23.9 \pm 1.152$ | $0.182 \pm 0.0151$ |
| | GraphSAGE | $22.2 \pm 1.703$ | $0.164 \pm 0.0129$ |
| | GAT | $22.9 \pm 0.596$ | $0.189 \pm 0.0042$ |
| | GIN | $26.4 \pm 1.240$ | $0.254 \pm 0.0143$ |
| | GraphMAE | $15.8 \pm 0.828$ | $0.083 \pm 0.0056$ |
| | GPS | $27.4 \pm 1.046$ | $0.268 \pm 0.0157$ |
| ViT embeddings | GCN | $16.0 \pm 0.801$ | $0.085 \pm 0.0050$ |
| | GraphSAGE | $15.8 \pm 0.980$ | $0.083 \pm 0.0064$ |
| | GAT | $20.5 \pm 3.941$ | $0.141 \pm 0.0490$ |
| | GIN | $45.4 \pm 0.670$ | $0.479 \pm 0.0059$ |
| | GraphMAE | $15.8 \pm 0.803$ | $0.083 \pm 0.0049$ |
| | GPS | $50.0 \pm 1.728$ | $0.530 \pm 0.0199$ |
| Image Pixels | FIMP-base (ours) | *$38.6 \pm 1.174$* | *$0.422 \pm 0.0170$* |
| | ViT | $\underline{56.5 \pm 3.187}$ | $0.597 \pm 0.0065$ |
| | FIMP-ViT (ours) | $\mathbf{63.2 \pm 0.764}$ | $\mathbf{0.684 \pm 0.0076}$ |

firming the performance benefits of leveraging non-textual foundation models in graph-based tasks. Furthermore, FIMP demonstrates zero-shot embedding capabilities on image networks that are on par with trained GNNs. This highlights the potential for zero-shot applications with pretrained non-textual foundation models on graphs despite them not natively taking graph structure as input.

There are several avenues for improvement upon our method, which we leave for future work. Currently, our evaluation of FIMP is limited to image and biological data. Protein design and social networks are promising areas of future research. Additionally, supporting multimodal graphs, heterogeneous graphs, and edge features would all expand the potential applications of FIMP. Finally, improving the scalability of FIMP to large graphs through strategies such as feature selection and efficient attention mechanisms beyond our usage of Flash Attention is an important future direction.

## 6 REPRODUCIBILITY STATEMENT

All datasets used in our experiments are publicly available, and are explained in section 4.1 and Appendix section A. Our experimental setup is explained in detail in Appendix section C. The foundation models used for our experiments are available through the Huggingface platform, and the architecture for FIMP is thoroughly discussed in section 3. We will release the full source code implementation of FIMP along with tutorial materials upon the paper's acceptance.

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

## A  Datasets (Extended)

**Spatial transcriptomics.** We use three publicly-available spatial transcriptomics datasets. The Slideseq-V2 spatial transcriptomics dataset (Stickels et al., 2021) is a mouse hippocampus dataset consisting of $41,786$ cells and $4,000$ genes, with $14$ different cell type classes. A second spatial dataset of human heart tissue was obtained from the 10X Genomics public spatial data repository, consisting of 4247 cells each with 36601 measured genes. A third spatial dataset, SeqFISH (Lohoff et al., 2020), consists of $15,000$ cells and $342$ genes taken from mouse embryo tissue sections. For all spatial transcriptomics datasets, we follow standard preprocessing and normalization procedures for RNA sequencing data, including count normalization and log transformation (Haque et al., 2017). For all datasets, we take the intersection of gene features which are present in the scGPT (Cui et al., 2023) pretrained foundation model, and split nodes into training, validation, and test sets with a 70/10/20 split. For graph adjacency information, we utilize the neighbor connectivity matrix present in each spatial transcriptomics dataset, which is derived from the original tissue section coordinates.

**Mapillary image dataset.** The Mapillary planet-scale image dataset (Antequera et al., 2020) is a dataset of 750,000 street-view images collected from over 170 countries around the world. Images are 1000-2000 pixels in height and width, originating from a variety of cameras and conditions depicting natural landscapes and buildings. Each image has a recorded latitude and longitude coordinate, forming a geographical proximity graph where each node represents a full image, connected to nearby image nodes if they are within 10 miles of one another. We evaluate FIMP on a geoguesser task, where the aim is to classify the country of origin based on the visual features of each image node and its neighborhood of nearby images. We train on 100,000 training images, and test on the predefined 10,000 test image set, with country labels determined for each image based on its latitude and longitude coordinates.

**fMRI brain activity recordings.** The UK Biobank dataset (Miller et al., 2016) comprises of 76,296 task-based and resting-state functional MRI (fMRI) recordings from 41,986 patients aged 40 to 69 years old. Recordings were acquired on a Siemens 3T scanner at 0.735s temporal resolution. All recordings went through standard preprocessing steps, including motion correction, normalization, temporal filtering, and ICA denoising (Salimi-Khorshidi et al., 2014; Abdallah, 2021). We parcellated the brain into 424 brain regions using the AAL-424 atlas (Nemati et al., 2020), yielding 424-dimensional scan sequences sampled at ª1 Hz. Finally, robust scaling was applied by subtracting the median and dividing by the interquartile range computed across subjects for each brain region. Our training set comprised of $60,000$ of the fMRI recordings, with the rest reserved for validation and test sets.

## B   NODE TOKENIZATION (EXTENDED)

The general formulation of node tokenization ($\tau$) becomes dataset-specific following tokenization schemes defined by foundation models on different data modalities. For instance, on datasets with input node feature vectors $\vec{x}_i \in \mathbb{R}^f$, such as a gene expression vector for a cell containing $f$ genes, we can see $X_i$ as an expanded feature vector with $c = 1$, and $\mathbf{W}$ as a projection of a scalar gene expression value into a $d$-dimensional vector embedding. The positional encoding $P$ would then represent a learned gene embedding $P \in \mathbb{R}^{f \times d}$, analogous to word embeddings in natural language. The concatenation operation in equation 5 would combine the expression value projection with its corresponding gene encoding, as in scGPT (Cui et al., 2023) and Geneformer (Theodoris et al., 2023).

For experiments on image datasets, $\tau$ is formulated as a patch encoding procedure following standard ViTs (Dosovitskiy et al., 2020), where an input image is divided into $f$ patches, each with $c$ pixels, that are embedded via a learned patch projector $\mathbf{W}$. Positional encoding $P$ is done through fixed 2D sinusoidal positional encoding which is concatenated with each patch embedding. For fMRI brain activity recordings, $\tau$ follows a spatiotemporal patching process as in the BrainLM foundation model (Ortega Caro et al., 2023), where for each brain region, segments of $c = 20$ signal timepoints are embedded via a learned projection $\mathbf{W}$. Spatial positional encoding is done through a learned projection of XYZ coordinates of each brain region, and temporal positional encoding is done using sinusoidal positional encoding.

## C   EXPERIMENTAL SETUP (EXTENDED)

The following section gives additional details about experimental setup across different datasets.

### C.1   IMAGE CLASSIFICATION

For image classification experiments, random 512x512 crops were taken from each image during training, with a 512x512 center crop taken at test time. Per-channel normalization was done on each image using statistics calculated across training images in the Mapillary image dataset. For FIMP and FIMP-ViT experiments, images were divided into 32x32 patches following the standard ViT patch encoding procedure (Dosovitskiy et al., 2020). For baseline GNNs, pixel values for each image were flattened and encoded using a learned projection.

### C.2   GENE EXPRESSION PREDICTION

For gene expression prediction experiments on spatial transcriptomics datasets, we limit the number of cells in each dataset to 5% of the original dataset size, leaving 1000 cells for the mouse hippocampus spatial dataset, and 200 cells for the human heart spatial dataset. This creates a challenging limited data setting for predicting gene expression values on each spatial dataset. We sample 50 nonzero expressed genes in each cell for all models and mask out 80% of the gene expression values, taking MSE loss against only masked out genes.

## C.3 fMRI RECORDING RECONSTRUCTION

In brain activity reconstruction experiments , we sample 320 consecutive timepoints from each fMRI recording, giving a recording of 424 brain regions with 320 timepoints of signal for each region. Each brain region is represented as 1 node in the graph, with node features being the 320 timepoints of signal. We segment the timepoints for each brain region into patches of 20 timepoints, and perform masked reconstruction of brain recording signals. For FIMP and variants of FIMP leveraging foundation models, masked patches are replaced with a mask token, and the signals are predicted back by the model. For baseline GNN models, node features comprise of the 320 timepoints of signal, and we explore three methods for replacing masked out patch values: i) replacing with random noise, ii) filling in with the mean value of the brain region, and iii) linearly interpolating between adjacent non-masked timepoint values. All models mask out $50\%$ of patches per each brain region, with mean squared error (MSE) taken against the original data.

## C.4 FOUNDATION MODELS

For experiments on single-cell datasets, the scGPT (Cui et al., 2023) whole-human checkpoint is incorporated for message creation in FIMP-scGPT, consisting of a 12-layer transformer with 54 million parameters. scGPT is pretrained using a masked gene expression prediction objective on over 33 million cells from a diverse array of human tissues and organs. The pretrained gene embedding table is also utilized from the pretrained scGPT checkpoint, representing pretrained knowledge about gene identities in transcriptomics datasets. Additionally, we also utilize the gene embeddings obtained by GenePT (Chen & Zou, 2023), which are GPT-3.5 embeddings of gene function descriptions based on biomedical literature, as another pretrained gene embedding experiment. For image classification, a standard ViT (Dosovitskiy et al., 2020) with 12 transformer layers and 86 million parameters is used as a message creator. The patch encoder from the ViT is also reused from the ViT embedding module. For experiments on fMRI brain recordings, the BrainLM (Ortega Caro et al., 2023) model was used, which consists of a Masked Autoencoder transformer with an 8-layer encoder and 4-layer decoder, totaling 26 million parameters.

## C.5 BASELINES

For both supervised and self-supervised tasks, we compare FIMP against popular message-passing GNN architectures, including GCN (Kipf & Welling, 2016), GraphSAGE (Hamilton et al., 2017), Graph Attention Networks (GATs) (Veličković et al., 2017), and Graph Isomorphism Networks (GINs) (Xu et al., 2018). We also compare FIMP against more recent GNN architectures, namely GraphMAE (Hou et al., 2022), a masked graph autoencoder model, and GPS Graph Transformer (Rampášek et al., 2022), a SOTA graph transformer framework. For supervised classification tasks, we additionally compare to the pretrained foundation model with no graph structure input.

# D RELATED WORKS

## D.1 ATTENTION-BASED GNNS AND GRAPH TRANSFORMERS

GATs (Veličković et al., 2017) first introduced the idea of attention-based GNN architectures, learning attention coefficients between neighboring nodes and performing message-passing with a weighted aggregation of neighboring node embeddings. Graph transformers sought to bring the performance and expressivity of the full transformer architecture into the graph domain by modeling graphs as a sequence of node embeddings that represented a fully-connected graph. Graph Transformer Networks (GTNs) (Yun et al., 2019) proposed the first graph transformer architecture, which could learn new graph structures and multi-hop connections. Graph-BERT (Zhang et al., 2020) proposed pretraining on subgraphs and finetuning for node classification and graph clustering tasks. Graph Transformer (Dwivedi & Bresson, 2020) proposed utilizing laplacian eigenvectors as positional encodings for node tokens. SAN (Kreuzer et al., 2021) improved upon it by introducing learnable spectral positional encodings, and Graphormer (Ying et al., 2021) further proposed spatial and centrality encodings for nodes to capture structural relation and node importance in graphs. GPS Graph Transformer (Rampášek et al., 2022) proposed a general framework for building expressive

graph transformers composed of positional and structural encodings, graph features, and GNN and attention layers.

In contrast to these works, **FIMP fundamentally redefines how nodes are represented by viewing each node as a sequence of feature tokens, similar to how transformer models handle input sequences, rather than as a single node embedding vector as in GATs and graph transformers**. This unique tokenization approach allows FIMP to compute cross-attention at the feature level between the token sequences of neighboring nodes, generating more informative messages that are passed between nodes in the graph. Unlike GATs and graph transformers, which focus on node-level attention, FIMP introduces feature-level attention for message creation. This makes FIMP the first approach to employ tokenized nodes for message-passing over graphs, leveraging the granularity of token interactions.

Additionally, FIMP's tokenization process aligns closely with the tokenization schemes of pretrained non-textual foundation models, minimizing distribution shift when repurposing these models to message-passing over graph-structured data. By integrating foundation models as message creators through this tokenization strategy, FIMP can effectively incorporate powerful pretrained representations in a way that traditional attention-based GNNs and graph transformers cannot.

### D.2    LLMs on Text-Attributed Graphs

More recent works have explored using Large Language Models (LLMs) in conjunction with LLMs on text-attributed graphs. GPT4Graph (Guo et al., 2023) evaluated LLM reasoning capabilities on graph reasoning tasks, establishing a benchmark of graph-related tasks for language models. Talk Like a Graph (Fatemi et al., 2023) and NLGraph (Wang et al., 2023) conducted similar studies exploring graph reasoning capabilities of LLMs, and released the GraphQA and NLGraph benchmark datasets, respectively. One-for-all (Liu et al., 2023) used LLMs as an encoding module for text-attributed graphs, and trained a unified GNN model to do node, edge, and graph-level classification using node-of-interest (NOI) subgraphs and prompt nodes. In contrast to these works, we focus on non-textual foundation models and graphs, which have not been explored extensively in graph-based tasks. Our work can be seen as a parallel work to LLM-based works on graphs, aiming to effectively leverage foundation models pretrained on other data domains besides natural language.

## E    Attention Visualizations

### E.1    Functional Region Attention in fMRI Recordings

During message passing on the fMRI recording graphs, FIMP generates cross-attention matrices during message-creation between feature tokens of neighboring brain regions in the K-nearest neighbors graph. We group the 424 brain voxels into 7 functional regions, namely the visual, sensorimotor, ventral salience, dorsal salience, central executive, default mode, and subcortical regions of the brain. Taking 100 unseen test set recordings, we extract attention matrices between all connected nodes, average the attention matrices across timepoints per node, and split patient recordings according to conditions such as Age and post-traumatic stress disorder (PTSD) score. We then average attention values across patient recordings with the same condition, and aggregate the node attention into the 7 functional regions, allowing us to examine differences in functional region attention between patients with different conditions.

In Figure 4A, the attention between functional regions is shown between patients below 65 years of age (left) and those above 65 (middle). The difference in attention between the two groups, as visualized on the rightmost plot, indicates that older patients tend to have higher attention between the dorsal salience regions and visual cortex regions. This follows previous literature that shows changes in dorsal pathways as people age (Yan et al., 2023). Furthermore, Figure 4B shows similar visualizations for patients with high and low PTSD scores, revealing higher attention between sensorimotor areas and central executive, and subcortical areas. This also follows previous literature on the somatosensory basis of PTSD, where arousal and higher-order capacities get affected (Kearney & Lanius, 2022). These patterns in attention reveal potential differences in functional region attention picked up by FIMP among patients of varying conditions.

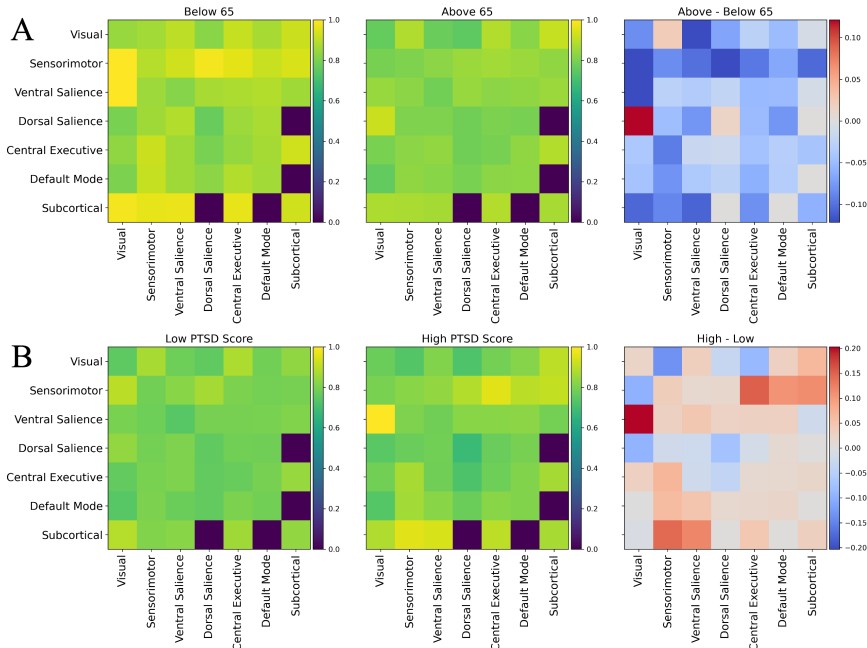

Figure 4: Visualizations of FIMP feature-level attention between different functional groups in the brain. (A) Averaged attention heatmaps between functional regions of the brain for different age populations, with the difference in attention by age group visualized on the right subplot. (B) Similar heatmaps visualized for post-traumatic stress disorder (PTSD) scores, highlighting differences in attention in patients with low vs high PTSD score.

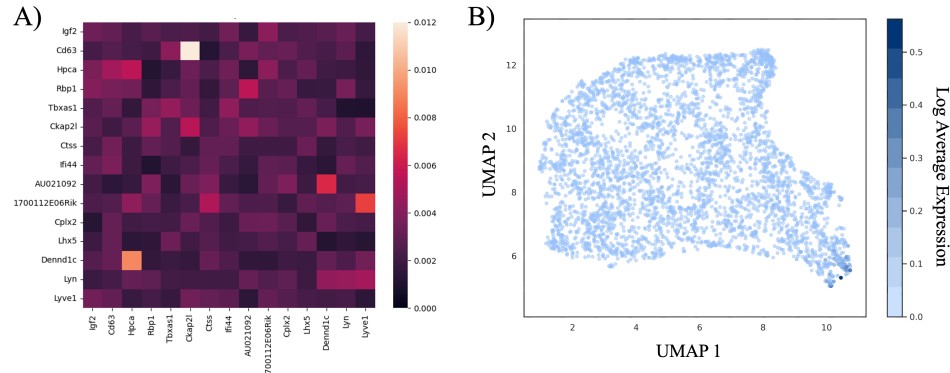

Figure 5: (A) Averaged attention between 15 genes across edges connecting neighboring astrocyte cells in the mouse hippocampus dataset. (B) UMAP of learned gene embeddings from FIMP, colored by average expression value of each gene across astrocyte cells.

### E.2    ATTENTION CASE STUDY 2: GENE INTERACTIONS IN SPATIAL TRANSCRIPTOMICS

In spatial transcriptomics datasets, each node corresponds to a cell which is represented by a set of expressed genes. Message-creation in FIMP provides cross-attention matrices representing interactions between genes of neighboring cells. Gene interactions receiving higher attention between nodes can highlight possible biological connections which can be avenues of potential further exploration in the data. For example, Figure 5A shows an averaged attention heatmap across all self-edges connecting astrocyte cells in a subgraph sampled from the mouse hippocampus dataset (Stickels et al., 2021). This astrocyte-astrocyte feature-level attention matrix identifies a key interaction between CD63, a member of the tetraspanin family of cell surface proteins, and CKAP2L, a mitotic

spindle protein controlling cellular division. Previous work has identified that CD63 may be either pro- or anti-tumorigenic, depending on tissue context (Dey et al., 2023). CD63 expression is also highly enriched in glioblastoma, a highly lethal malignancy of the astrocytes, and may play a role in progression of these cancers (Aaberg-Jessen et al., 2018). This hints that CD63 may play an important role in controlling cellular division through astrocyte-astrocyte cellular communication, which may represent an exciting new target for antitumoral agents.

Figure 5B shows a UMAP embedding of the gene embeddings learned by FIMP-base during masked gene expression prediction training. Each gene is colored by its average expression value across all astrocyte cells in the mouse hippocampus dataset. We see that the learned embeddings form distinct structures during training, and that highly-expressed genes for astrocytes are clustered together in one region in the bottom-right. We hypothesize that this ability to learn gene vectors in embedding space and contextualize them for different cell types allows FIMP to outperform other methods in gene expression prediction tasks.

## F  TRAINING TIME

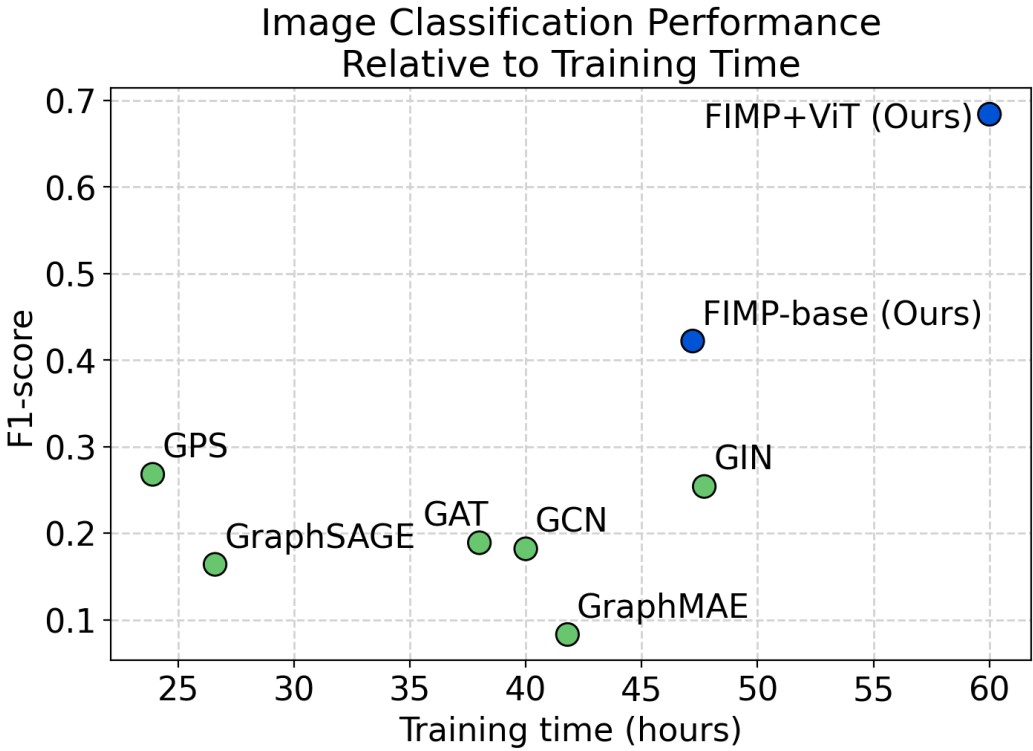

Figure 6: This figure illustrates the relationship between training time (in hours) and image classification performance for FIMP compared with other GNN baseline models. It highlights how FIMP, when leveraging a ViT model, improves performance by 63% over FIMP-base while only adding 27% more training time.

We measure the training time of various GNN baseline models compared to variants of FIMP with and without foundation model layers on the image classification task, to analyze the performance gained versus additional compute overhead required. Figure 6 demonstrates that with a small increase in training time, FIMP-base and FIMP-ViT are able to achieve significantly higher performance on the image classification task compared to GNN baseline models. This highlights that the additional compute when applying pretrained foundation models for message-passing in graph settings can yield improved performance at a small cost in increased training time.

