# OpenReview forum: "FIMP: Foundation Model-Informed Message Passing for Graph Neural Networks"
_ICLR.cc/2025/Conference — Submitted to ICLR 2025_

### Official Review · Reviewer_uKm7 · 2024-11-02

**Soundness:** 3
**Presentation:** 2
**Contribution:** 2
**Rating:** 5
**Confidence:** 4

**Summary:**

The paper introduces FIMP, a framework that adapts pretrained non-textual foundation models for graph-based tasks. FIMP repurposes existing pretrained non-textual foundation models for message-passing on graphs. FIMP is evaluated across diverse domains, such as image classification, spatial transcriptomics, and fMRI brain activity recordings, utilizing state-of-the-art foundation models like ViTs, scGPT, and BrainLM. It demonstrates competitive performance in a zero-shot setting for embedding image networks, achieving notable results without task-specific retraining.

**Strengths:**

1. The concept of leveraging pretrained foundation models for message passing in GNNs is innovative. This approach shows how pretrained knowledge can inform finer-grained interactions in graph structures.
2. The suggested method achieves improvements over the discussed baselines.

**Weaknesses:**

1. The paper is not easy to follow, with several concepts and sections lacking clarity (see questions below).
2. While the idea of integrating pretrained foundation models with MPNNs is interesting, the proposed approach is overly simplistic (a straightforward approach -- if I understood correctly –- see questions below). Simplicity in methodology is not inherently problematic in my opinion; however, the combination of simplicity with a lack of a clear motivation for addressing the problem raises concerns — see Weakness 4 for further details.
3. The experimental results are not entirely convincing, as the baselines used for comparison (e.g., GCN, GAT) are somewhat outdated. It would strengthen the paper to include comparisons with more recent state-of-the-art graph learning (e.g.,  GRIT [1]) methods that could offer a more rigorous benchmark for evaluating FIMP's performance.
4. In my opinion, the paper lacks sufficient motivation. The tasks presented appear to be crafted primarily to emphasize FIMP’s strengths, rather than addressing meaningful, real-world applications. If I understand correctly, the authors construct a graph from publicly available datasets (e.g., the Mapillary image dataset). For example, line 173 states: “images form a geographical proximity graph.” However, this implies that the method is evaluated exclusively on datasets where the graphs were constructed by the authors, and these graphs are not publicly accessible.
To strengthen the paper, the authors can either (1) provide a compelling rationale for how such a setting could arise naturally in real-world scenarios, or (2) identify and utilize publicly available datasets that align with their setup.

**Questions:**

1. Lines 191-193: "By aligning the tokenization in FIMP with the tokenization scheme of pretrained foundation models, we reduce distribution shift in token representation when applying these models to graph-structured data." Does this mean that you are directly adopting the tokenization strategy of the pretrained foundation model? Clarification on how closely you follow the original tokenization would be helpful.
2. In Algorithm 1, which of the weights are learned from scratch, and which are (possibly) borrowed from the pretrained foundation model? A clearer distinction between learned and reused weights would improve understanding.
3. How does FIMP-base specifically leverage the knowledge embedded in a foundation model? From what I understand, given lines 251-253: “In its base formulation, cross-attention message passing can be done with a simple cross-attention mechanism which is learned from scratch during training. We denote this base version of our architecture as FIMP-base in our experiments.”, the knowledge transfer in this case occurs primarily through the tokenization process.
4. On the FIMP versions that rely on a foundation model’s weights in the cross attention, It seems unclear how effective using the pretrained weights of a foundation model would be, given that cross-node attention differs from what the foundation model encountered during training (e.g., ViTs typically see only single images per training step). Could you elaborate on how the cross-node attention mechanism adapts to this discrepancy? Or did it just appear to work well in practice?

**References**:

[1] Graph Inductive Biases in Transformers without Message Passing. Ma et. al. 2023

---

> ### Author Response · Authors · 2024-11-20
> **Rebuttal, Part 1**
>
> Thank you for your detailed review and for identifying strengths and areas for improvement in our submission. We appreciate your balanced evaluation and the opportunity to address your concerns. Below, we provide clarifications and additional context in response to your feedback:
>
> ### **Clarity and Simplicity of the Approach**
> While we acknowledge that FIMP leverages a straightforward integration of foundation models with MPNNs, we believe its simplicity is a strength rather than a limitation. This simplicity allows for:
> 1. **Effective Generalization:** By repurposing pretrained foundation models through tokenization and cross-node attention, FIMP achieves significant performance gains across different graph domains without requiring complex architectural modifications. Our main goal is to align pretrained foundation models to work on graph-structured inputs, allowing pretrained representations from foundation models on unstructured data to benefit downstream graph-based applications.
> 2. **Interpretability:** The tokenization and cross-attention mechanisms provide interpretable feature-level interactions, as detailed in Appendix E.
>
> ### **Use of Outdated Baselines**
> We appreciate the recommendation to include comparisons with more recent state-of-the-art graph learning methods!. While we focused on widely-used baselines (e.g., GCN, GAT) and strong graph transformer baselines (GPS Graph Transformer) for comparability with prior works, we agree that incorporating modern baselines such as GRIT [1] would strengthen the experimental evaluation. We are working to add more updated GNN baselines, which we hope to complete during this discussion period.
>
> ### **Motivation and Real-World Relevance**
> 1. **Choice of Tasks and Motivation**
>     - The tasks we selected are designed to benchmark FIMP in domains where node features are rich, complex, and high-dimensional, such as cell expression profiles, images, and fMRI recordings. These domains particularly benefit from the integration of pretrained foundation models, which excel at learning general representations from large-scale unstructured data. By leveraging these pretrained representations, FIMP achieves significant performance improvements over traditional GNN baselines, as shown in our experiments. We argue that such tasks are not only appropriate benchmarks for evaluating FIMP’s strengths but also highlight its utility in handling complex, real-world data scenarios.
> 2. **Real-World Applicability**
>     - For all spatial genomics datasets used in our experiments, the graph structure comes as a part of the publicly available data. As noted in the introduction, there are large amounts of unstructured cell expression data publicly available compared to spatial genomic data which preserves the graph structure inherent in tissues. We argue that utilizing foundation models pretrained on unstructured expression data (e.g., scGPT) for spatial genomics tasks provides significant real-world advantage over traditional methods which do not leverage pretraining on unstructured data in spatial genomics tasks. We believe this application demonstrates the meaningful, real-world impact of FIMP.
>     - For the Mapillary image dataset and fMRI recording datasets, the latitude/longitude coordinates for images and the voxel positions for fMRI recordings are publicly available as part of the datasets as well. The graph constructions we performed (e.g., K-nearest neighbors for fMRI voxels or proximity graphs for Mapillary images) are simple preprocessing steps that anyone can reproduce using the provided dataset features. We argue that these settings represent real-world graph-structured data where foundation models pretrained on unstructured data can be leveraged.
>
> To further ensure reproducibility, we plan to release all datasets and code after the review process, including the graph structures we used and the code for reproducing the preprocessing steps.
>
> ### **(Q1) Tokenization Alignment with Foundation Models**
> Great question! For spatial genomics datasets, we directly adopt the gene tokenization encoder from scGPT to encode cell expression profiles. This keeps the gene feature representation per cell consistent with what scGPT saw during pretraining, ensuring minimal distribution shift. For image data and fMRI recordings, we again closely follow the patch-based tokenization of ViT and BrainLM, respectively, dividing images or fMRI recordings into fixed-size patches. Appendix Section B further describes how tokenization schemes are adapted for each dataset. We will further expand this section to include more details explaining the original tokenization algorithm of each foundation model used in experiments.

---

> ### Author Response · Authors · 2024-11-20
> **Rebuttal, Part 2**
>
> ### **(Q2) Learned vs. Reused Weights in Algorithm 1**
> Another great question! Further clarification on algorithm 1 is given below:
> 1. **Learned Weights:** When trained completely from scratch (FIMP-base), all attention weights ($W_Q$, $W_K$, $W_V$) are initialized randomly and learned during training. FIMP-base therefore provides performance measures based solely on the improved tokenization of input graph data as well as cross attention message passing.
> 2. **Reused Weights:** When using a pretrained foundation model on the graph-based task (FIMP-scGPT, FIMP-ViT, FIMP-BrainLM), the pretrained self-attention layers of the foundation model attention are used in place of $W_Q$, $W_K$, and $W_V$, and are finetuned along with the rest of the model for the task.
> We will revise the description of Algorithm 1 to distinguish between learned and reused weights and add this clarification in Section 3.3.
>
> ### **(Q3) Knowledge Transfer in FIMP-Base**
> We would like to clarify that FIMP-base represents a version of the FIMP architecture that does not leverage pretrained foundation models. FIMP-base follows the same tokenization procedure as FIMP + foundation models for each task, however all encoders and weights of FIMP-base are trained from scratch. This is done mainly to demonstrate how the improved tokenization procedure and cross-attention message passing improve performance over GNN baselines even when trained completely from scratch. The FIMP + foundation model variants then go further and leverage pretrained foundation models in the message-passing framework, demonstrating that the pretrained representations further improves performance on each task.
>
> We hope that this clarifies the distinction between FIMP-base and FIMP + foundation model variants. We will clarify this distinction in Sections 3.2–3.3 in updated versions of the manuscript.
>
> ### **(Q4) Adapting Pretrained Weights for Cross-Node Attention**
>
> Great question! We would like to take this opportunity to explain in more detail how the cross-node attention works. The foundation models contain pretrained transformer layers, which each contain self-attention layers. These self-attention layers are pretrained to do query-key-value (QKV) attention on input sequences. Specifically, in each self-attention layer, the input feature sequence is projected into query tokens, key tokens, and value tokens by the $W_Q$, $W_K$, and $W_V$ matrices, respectively. The attention matrix is computed through QK, and the attention matrices are then multiplied with the value vectors to form the output of attention.
>
> Our main observation is that cross-attention can be formulated with the same layers if the attention is done between two sequences of features rather than one: Q representing the destination node feature sequence, and K and V representing the source node feature sequence. This represents a shift from the layer performing self-attention on unstructured data to performing cross-attention between neighboring nodes in a graph. We find that this empirically works well, and allows the cross-attention between nodes to leverage the pretrained weights which learned to compute self-attention on large amounts of unstructured data.
>
> We hope these responses clarify the motivation, methodology, and contributions of FIMP. We are conducting additional experiments to address your points which we hope to add by the end of the rebuttal period. We respectfully request that you consider these clarifications when reevaluating your score.

---

> > ### Comment · Reviewer_uKm7 · 2024-11-23
> >
> > I appreciate the authors' thorough rebuttal and the detailed clarifications provided in response to my questions. I have also read the other rebuttals.
> >
> > However, as I previously noted, "the combination of simplicity with a lack of clear motivation for addressing the problem raises concerns." Despite the explanations offered, I still find the motivation presented in the rebuttal insufficient. Consequently, my primary concern regarding the simplicity of the approach remains -- I find the proposed method overly simplistic and incremental.
> >
> > I will maintain my current score.

---

> > > ### Author Response · Authors · 2024-12-01
> > > **Additional GNN Baselines, Part 1**
> > >
> > > Thank you again for highlighting potential GNN architectures for us to compare against in our benchmarks. We have now incorporated GRIT as a baseline into two of our experiments: mouse hippocampus cell classification and Mapillary image classification. Below, we summarize the benchmarking results:
> > >
> > > ### Table 1. Results on Mapillary image classification task.
> > > Method | Accuracy | F1-Score
> > > -----|-----|-----
> > > GCN | 23.9 ± 1.152 | 0.182 ± 0.0151
> > > GraphSAGE | 22.2 ± 1.703 | 0.164 ± 0.0129
> > > GAT | 22.9 ± 0.596 | 0.189 ± 0.0042
> > > GIN | 26.4 ± 1.240 | 0.254 ± 0.0143
> > > GraphMAE | 15.8 ± 0.828 | 0.083 ± 0.0056
> > > GPS | 27.4 ± 1.046 | 0.268 ± 0.0157
> > > **GRIT** | 19.7 ± 0.863 | 0.224 ± 0.00785
> > > FIMP-base | 38.6 ± 1.174 | 0.422 ± 0.0170
> > > ViT | 56.5 ± 3.187 | 0.597 ± 0.0065
> > > FIMP-ViT | **63.2 ± 0.764** | **0.684 ± 0.0076**
> > >
> > > ### Table 2. Results on mouse hippocampus cell classification task.
> > > Method | Accuracy | F1-score
> > > -----|-----|-----
> > > GCN | 47.59 ± 3.788 | 0.445 ± 0.050
> > > GraphSAGE | 51.81 ± 3.229 | 0.495 ± 0.036
> > > GAT | 46.21 ± 3.110 | 0.442 ± 0.031
> > > GIN | 52.71 ± 0.421 | 0.507 ± 0.008
> > > GPS | 52.89 ± 1.176 | 0.510 ± 0.008
> > > **GRIT** | **56.65 ± 3.961** | **0.558 ± 0.037**
> > > FIMP-base | 49.04 ± 1.215 | 0.464 ± 0.019
> > > scGPT | 53.50 ± 0.424 | 0.518 ± 0.005
> > > FIMP-scGPT | **57.05 ± 1.393** | **0.554 ± 0.004**
> > >
> > > On the mouse hippocampus cell classification task, GRIT performs well, on par with or slightly below FIMP in accuracy and F1-score. This demonstrates that GRIT is indeed a strong baseline for cell type prediction tasks with simpler node features.
> > >
> > > On the Mapillary image classification task, GRIT performs significantly worse than FIMP, similar to other GNN baselines. We attribute this to the fact that GRIT, like traditional GNNs, encodes nodes as single embedding vectors, which is less effective for image data. In contrast, FIMP leverages Vision Transformer (ViT) models for image-based tasks, enabling it to better encode high-dimensional, feature-rich image data.
> > >
> > > These results further highlight FIMP's ability to align pretrained foundation models with graph-based tasks, particularly in domains with complex node features. While GRIT is a robust GNN baseline, our findings suggest that FIMP’s tokenization and cross-attention mechanisms provide a clear advantage for tasks where node features are high-dimensional and benefit from pretrained representations.

---

> ### Author Response · Authors · 2024-12-01
> **Clarification on Motivation and Simplicity, Part 2**
>
> We would also like to address the points raised regarding the motivation for FIMP and concerns about its simplicity. There are numerous real-world datasets with rich, information-dense features that can benefit from graph representation learning. For instance, Relational Deep Learning (RDL) [1] presents a deep learning approach for relational databases, highlighting opportunities for learning on structured data stored in such systems. As noted in our response to Reviewer **Gmtx**, simpler GNN models perform well on graph datasets with straightforward node features [2]. However, they struggle on datasets with more complex features, such as gene expression data in spatial genomics, spatiotemporal fMRI signals, and pixel-based image data. Across these domains, FIMP consistently outperformed all GNN baselines, demonstrating the meaningful impact of aligning pretrained foundation models to encode node entities and perform message passing effectively. Furthermore, FIMP uniquely enables zero-shot capabilities, as demonstrated on the Mapillary dataset. This allows FIMP to apply non-textual foundation models to graph datasets without any additional training, something that neither GNN baselines nor recent state-of-the-art methods can achieve.
>
> Finally, we would like to reemphasize the motivation behind integrating foundation models into our cross-node attention message passing framework. FIMP is intended to bridge the gap between non-textual foundation models and graph-based tasks. Its simple yet effective design enables easy adoption across various domains, as demonstrated in our experiments. To our knowledge, FIMP is the first general framework to showcase the advantages of non-textual foundation models in graph settings. We believe this represents a significant contribution, paving the way for future applications of foundation models in graph learning, including zero-shot scenarios.
>
> We will update our manuscript in future revisions to include these points and additional baselines, and with more time, we will incorporate GRIT into all other benchmarking tasks as well. We hope that these additional responses clarify why we believe FIMP represents a strong contribution to the field, and we kindly ask you to reconsider your score. Thank you for your continued engagement during the discussion phase.
>
> References
> [1] Fey, Matthias and Hu, Weihua and Huang, Kexin and Lenssen, Jan Eric and Ranjan, Rishabh and Robinson, Joshua and Ying, Rex and You, Jiaxuan and Leskovec, Jure. RelBench: A Benchmark for Deep Learning on Relational Databases
> [2] Yuankai Luo, Lei Shi, Xiao-Ming Wu. Classic GNNs are Strong Baselines: Reassessing GNNs for Node Classification

---

> ### Author Response · Authors · 2024-12-02
> **Additional GNN Baseline Results**
>
> In addition to our previous inclusion of the GRIT baseline, we have now also evaluated Graph Substructure Network (GSN) [1], another state-of-the-art GNN baseline, on the masked gene expression prediction task. Below, we present the results for GSN and other models on the Slideseq-V2 mouse hippocampus dataset:
>
> Method | MSE | R^2
> -----|-----|-----
> GCN | 0.0211 ± 0.0018 | 0.0236 ± 0.0457
> GraphSAGE | 0.0181 ± 0.0012 | 0.1853 ± 0.0306
> GAT | 0.0201 ± 0.0008 | 0.0905 ± 0.0233
> GIN | 0.0175 ± 0.0009 | 0.1707 ± 0.0424
> GraphMAE | 0.0178 ± 0.0006 | 0.1538 ± 0.0254
> GPS | 0.0149 ± 0.0012 | 0.2977 ± 0.0308
> scGPT | 0.0169 ± 0.0007 | 0.2087 ± 0.0191
> **GSN** | 0.0160 ± 0.0013 | 0.3156 ± 0.0369
> FIMP-base | 0.0134 ± 0.0009 | 0.3815 ± 0.0226
> FIMP-GenePT | 0.0129 ± 0.0005 | 0.4058 ± 0.0302
> FIMP-scGPT | **0.0119 ± 0.0008** | **0.4612 ± 0.0029**
>
> While time constraints during the rebuttal period prevented us from running GSN across multiple tasks, these results demonstrate that FIMP outperforms SOTA GNNs, including GSN, on this gene expression prediction task. The performance gap highlights FIMP’s ability to handle complex graph datasets and its effectiveness in leveraging pretrained foundation models for graph-based tasks.
>
> We kindly ask the reviewer to consider these additional experimental results in their evaluation of our work. Thank you for your continued engagement during the rebuttal process.
>
> [1] Bouritsas, Giorgos, et al. "Improving graph neural network expressivity via subgraph isomorphism counting." IEEE Transactions on Pattern Analysis and Machine Intelligence 45.1 (2022): 657-668.

---

### Official Review · Reviewer_Gmtx · 2024-11-03

**Soundness:** 3
**Presentation:** 3
**Contribution:** 2
**Rating:** 5
**Confidence:** 3

**Summary:**

This paper introduces a message-passing framework that leverages pretrained non-textual foundation models for graph-based tasks. The proposed method, FIMP, aligns node tokenization in Graph Neural Networks with tokenization schemes used by foundation models, allowing for more granular feature-level interactions during message passing. The authors demonstrate FIMP's effectiveness across diverse domains, including image networks, single-cell RNA sequencing, and fMRI brain activity recordings, showing improvements over strong baselines and highlighting the potential of repurposing non-textual foundation models for graph tasks.

**Strengths:**

1. The expression is very fluent. Readers can easily understand the content of the work.

1. The study of graph foundation models and large graph models is very meaningful.

1. The paper demostrates FIMP outperforms traditonal GNN models on image networks, single-cell RNA sequencing, and fMRI brain activity recordings. FIMP can effectively leverage SOTA foundation models in graph tasks.

**Weaknesses:**

1. The algorithm proposed in this paper appears to be incremental. Although the paper discusses the differences from works like GAT, in essence, it merely transforms node features from vectors($x_v \in \mathbb{R}^d$) to matrices($x_v \in \mathbb{R}^{f \times d}$).

2. The current experiments are limited to image networks, single-cell RNA sequencing, and fMRI brain activity. It is unclear whether this method is effective in other domains, especially traditional graph datasets. I hope to see experiments conducted on larger practical datasets like OGB.

3. As shown in Table 1, the gene expression prediction results on the mouse hippocampus and human heart spatial transcriptomics datasets, FIMP + scGPT performs much better than FIMP-base and FIMP + ViT. This indicates that to unleash the potential of the FIMP algorithm, a suitable and powerful foundation model is needed. However, in many scenarios, there may not be powerful foundation models available. Pretraining a suitable foundation model from scratch is also expensive and challenging. This limits the practical application of FIMP.

**Questions:**

The practical application of traditional GNNs is hampered by a significant computational burden. As shown in Figure 6, the time cost of FIMP is twice that of traditional GNNs. This concerns me about the practical value of FIMP. I would like to see a comparison of F1-Score and consumed time between FIMP + ViT and the ViT model alone on image classification dataset. I also hope to see the training time expenditure of FIMP on larger graph datasets.

In summary, if the authors can address my concerns, I would be open to increase my score.

---

> ### Author Response · Authors · 2024-11-20
> **Rebuttal, Part 1**
>
> Thank you for your review and for highlighting aspects of our work that could be improved. We appreciate the opportunity to address your concerns and clarify the contributions and broader impact of FIMP. We respond to each comment in detail:
>
> ### **Novelty of the FIMP algorithm**
> We agree that attention-based Graph Neural Networks (GNNs) and transformers are well-studied in graph learning, however we argue that FIMP introduces significant innovations that distinguish it from prior methods:
> 1. **Feature-Level Message Passing:** Unlike traditional GNNs, including attention-based ones like GATs, which use scalar attention at the node level, FIMP tokenizes nodes into sequences of feature vectors and applies cross-node attention between feature tokens. This enables a finer-grained representation of node interactions, analogous to how transformers revolutionized NLP by moving from bag-of-words models to sequence-based architectures. This represented a major shift in data representation and generalizability in NLP, and we believe that similar gains in performance are attainable in graph settings based on our results.
> 2. **Foundation Model Integration:** FIMP uniquely integrates pretrained non-textual foundation models as message creation modules in graph settings. This allows us to leverage representations learned on vast amounts of unstructured data, enabling performance improvements and zero-shot capabilities that are not possible with traditional GNNs or standalone foundation models that do not operate on graphs.
> 3. **Zero-Shot Capabilities:** By repurposing pretrained foundation models, FIMP enables zero-shot applications on graph-structured data—a capability not achievable with traditional GNNs. This is particularly impactful in domains where labeled graph data is scarce and unstructured data is available for pretraining, such as single-cell genomics.
>
> A more detailed comparison of FIMP with traditional GNNs is provided in Section 2.2 and Appendix D.
>
> ### **Evaluation on Traditional Graph Datasets**
> We deliberately focused our evaluation on domains where node features are rich and complex, as these settings highlight the advantages of FIMP’s feature-level tokenization and pretrained foundation model integration. Traditional graph benchmarks like OGB often have simpler node features (e.g., categorical or numeric vectors), where the benefits of foundation models are less pronounced, and simpler GNNs already achieve strong performance. Specifically, we focused on graph-structured data that was not bag-of-words or bag-of-features, since the preprocessing of these datasets loses information which foundation models can utilize, similar to how Large Language Models model natural language at a deeper level than term frequency or other bag-of-words based methods.
>
> We will revise our limitations section to emphasize that FIMP is not best for every graph scenario. FIMP provides larger benefits when node entities have rich features that are not easily modeled by traditional GNNs.
>
> ### **Dependence on Powerful Foundation Models**
> We agree with the reviewer that FIMP’s reliance on powerful pretrained foundation models might limit its practicality in domains lacking such models. This is an important consideration, however we would like to highlight the following:
> 1. **Performance Without Pretrained Models:** FIMP-base, which does not use pretrained foundation model weights, still outperforms traditional GNNs on several tasks. This demonstrates that FIMP’s tokenization and cross-node feature-level attention are inherently advantageous, even without pretrained weights.
> 2. **Foundation Models Across Domains:** While foundation models are not yet available for every domain, their prevalence is rapidly increasing across fields such as biology, neuroscience, and vision. As highlighted by recent surveys [1], the development of domain-specific foundation models is accelerating, providing opportunities for methods like FIMP to bridge the gap between unstructured and graph-structured data as Reviewer ZuVo noted.
>
>
> **References**
> [1] Zhou, Ce, et al. "A comprehensive survey on pretrained foundation models: A history from bert to chatgpt." arXiv preprint arXiv:2302.09419 (2023).

---

> ### Author Response · Authors · 2024-11-20
> **Rebuttal, Part 2**
>
> ### **Computational Efficiency**
>
> We agree with the reviewer that the computational burden of leveraging foundation models in graphs is an important consideration. While FIMP’s use of foundation models introduces additional computational costs, we argue that this trade-off is justified in domains with complex node features. In the Mapillary dataset, for example, traditional GNNs fail to encode the details contained in the high-dimensional images. FIMP’s integration of pretrained ViTs enables significant performance gains, as shown in Table 3.
>
> Regarding larger graph datasets, the Mapillary dataset is already a large dataset with over 100,000 images in the network. We show that by training on subgraphs sampled from the entire graph, FIMP-ViT can still effectively learn and generalize on the image classification task. In domains where computational resources are limited, we note that FIMP-base provides a more efficient alternative while still outperforming traditional GNNs.
>
> Regarding the runtime comparison of FIMP + ViT and ViT itself on the image classification task, we provide the comparison in the table below:
> | Model | Runtime | F1-score |
> | ---- | ---- | ---- |
> | ViT | 17.6 hours | 0.597 |
> |FIMP + ViT | 60.0 hours | 0.684 |
>
> We note here that the ViT model has less training time because it does not take any graph structure as input, and only processes a single image on a forward pass for classification. FIMP + ViT in comparison processes input subgraphs of images, which takes more time but achieves significantly higher F1-score on the image classification task since it considers a neighborhood of images by geographical proximity.
>
> ### **Broader Contributions**
> We also want to reiterate the broader impact of FIMP. By bridging the gap between graph-based tasks and pretrained non-textual foundation models, FIMP introduces a new paradigm for leveraging large-scale pretraining in graph learning. Beyond its technical contributions, FIMP opens up exciting possibilities for zero-shot applications and interpretable graph-based learning.
>
> We hope these responses address your concerns and clarify the unique contributions of FIMP. We respectfully request that you reconsider your evaluation in light of this additional context, and we welcome any further questions or feedback you may have.

---

### Official Review · Reviewer_YLzx · 2024-11-08

**Soundness:** 2
**Presentation:** 3
**Contribution:** 1
**Rating:** 3
**Confidence:** 4

**Summary:**

This paper incorporates pre-trained LM for cross-node message creation for a non-textual graph neural network. To this end, it tokenizes each node's content into a sequence and employs self-attention between sequences from pairs of nodes. Evaluations on genomics, brain, and street-view images demonstrate its effectiveness.

**Strengths:**

1. The writing and organization are good. The presentation is easy to follow.2.
2. The proposed methods can be applied to many fields.
3. Experimental evaluations from different fields demonstrate its effectiveness.

**Weaknesses:**

1. The novelty is very limited. The proposed method is the extension of attention in GAT to non-textual node content. This aims at jointly considering source and target nodes for message passing. However, this is not novel.
2. The proposed method lacks clear motivation. It is direct for non-textual node content. Thus, the significance is weak. It is not clear why the cross-node attention can improve the performance.
3. The evaluations are not convincing. Only the GNN baselines are compared. Besides, it is not clear how the node attribute is constructed for GNNs. How about using the embedding from LM as the node’s attribute?
4. Figure 3 lacks a description in the main body.

**Questions:**

See weaknesses.

---

> ### Author Response · Authors · 2024-11-19
> **Rebuttal, Part 1**
>
> Thank you for taking the time to review our submission, we appreciate your constructive feedback. Below, we respond to your comments and aim to clarify potential misunderstandings about our work:
>
> ### **Novelty of FIMP Compared to GATs**
> We respectfully disagree with the characterization that FIMP is an extension of GATs to non-textual node content, and would like to clarify the key distinctions between FIMP and GATs:
> 1. **Node Representation:** Traditional GNNs, including GATs, represent nodes as single embedding vectors, while FIMP tokenizes nodes into sequences of feature vectors (tokens). This tokenization aligns with foundation model architectures, enabling feature-level interactions that GATs cannot achieve.
> 2. **Message-Passing Mechanism:** GATs use scalar attention coefficients to aggregate single-node embeddings. In contrast, FIMP employs cross-node attention at the feature level between neighboring nodes, leveraging pretrained self-attention layers from foundation models as processing layers. This allows FIMP to capture fine-grained inter-node dependencies, enhancing expressivity and interpretability.
> 3. **Foundation Model Integration:** Unlike GATs, FIMP directly incorporates pretrained non-textual foundation models into its message-passing framework, enabling zero-shot applications on graphs and leveraging pretrained representations from domains such as images, genomics, and neuroscience. These capabilities are not achievable with GATs.
>
> Sections 2.2 and Appendix D further emphasize the differences between FIMP and existing attention-based GNNs. We argue that traditional GNNs such as GAT cannot integrate foundation models in the same way that FIMP does, due to representational differences of node entities.
>
> ### **Motivation for Cross-Node Attention**
> We appreciate your suggestion to clarify the motivation behind cross-node attention and its impact on performance. Cross-node attention between feature sequences of neighboring nodes enables feature-level interactions, offering several advantages:
> 1. **Enhanced Interactions:** By tokenizing nodes into sequences, FIMP models $O(F^2 \cdot E)$ feature-level relationships, where $F$ is the number of features per node and $E$ is the number of edges. This considers significantly more interactions than traditional GNNs, which only model $O(E)$ node-to-node relationships. This can benefit graph domains with complex node features such as genomics where genes of neighboring cells may affect cell-cell interactions.
> 2. **Improved Representational Power:** Similar to how transformers outperform bag-of-words models in NLP by tokenizing input text, FIMP’s token-based representation allows for more contextualization of node features, leading to better performance across multiple tasks.
> 3. **Interpretability:** Attention coefficients between features of neighboring nodes can be visualized, providing insights into which node features contribute most to interactions during message-passing.
>
> ### **Comparisons With Baselines**
> FIMP is compared against both traditional GNN baselines and the foundation models themselves (e.g., scGPT, ViT, BrainLM) on each task, which represents a strong non-GNN baseline in each of their respective domains (single-cell genomics, images, and fMRI recordings, respectively). This demonstrates that FIMP effectively integrates pretrained models into a graph-based framework, outperforming both types of baselines.
>
> ### **Node Attribute Construction**
> For GNN baselines, we use raw node features from each dataset, consistent with standard preprocessing steps in GNN literature. For example, in the spatial transcriptomics datasets, node features are preprocessed gene expression vectors for each cell. For FIMP, each node entity is tokenized according to the methodology detailed in Section 3.1. A more detailed description of how node attributes are constructed is provided in Appendix C (Experimental Setup Extended).
>
> ### **Description of Figure 3**
> Thank you for pointing out the lack of description for Figure 3! We will add a detailed explanation of this figure in the Results section to ensure clarity. Specifically, Figure 3 summarizes FIMP’s performance gains over GNN baselines across tasks, demonstrating its effectiveness in diverse graph-based settings.

---

> ### Author Response · Authors · 2024-11-19
> **Rebuttal, Part 2**
>
> ### **Broader Contributions**
> We also wish to reiterate the broader contributions of FIMP, which may have been unclear. Beyond its technical novelty, FIMP introduces a framework that bridges the gap between non-textual foundation models and graph-structured data. By enabling these pretrained models to operate on graphs, FIMP unlocks new capabilities such as zero-shot graph tasks and interpretable feature-level message-passing
>
> We hope these clarifications address your concerns and provide a clearer picture of the novelty, motivation, and contributions of FIMP. We respectfully request that you reconsider your evaluation in light of this additional context, and we are happy to further discuss any remaining questions. Thank you again for your time and feedback.

---

> ### Author Response · Authors · 2024-12-01
> **Request for Further Feedback**
>
> Thank you again for reviewing our paper and providing your initial comments. We recognize your concerns regarding the novelty and motivation of FIMP, and we have taken significant steps during the rebuttal process to clarify these aspects and address the issues raised.
>
> In particular, we have provided detailed comparisons between FIMP and other baselines, incorporated new experiments (e.g., benchmarking GRIT on key tasks), and further expanded on the key distinctions between FIMP and traditional GNN architectures like GATs. These updates are described in our responses to other reviewers, and we encourage you to review these responses which highlight the additional experiments and insights we have added to strengthen the paper.
>
> In particular, we would like to emphasize three key points about FIMP that we believe represent meaningful contributions of our work to the field:
> 1. **FIMP is a natural choice for graph representation learning in settings with rich node features.** FIMP is particularly well-suited for graph representation learning in settings with rich, complex node features. As noted in our response to Reviewer **uKm7**, traditional GNNs perform strongly on graphs with simple node features. However, in graphs with more complex and information-rich features, stronger models are needed to effectively encode and process this information. FIMP leverages pretrained foundation models, which incorporate representations learned from unstructured data, to address this need. In our experiments, FIMP consistently outperforms even state-of-the-art methods such as GRIT [1] across multiple tasks. We attribute this superior performance to FIMP’s ability to align pretrained representations with graph-structured inputs, demonstrating its effectiveness in handling complex graph datasets.
> 2. **FIMP establishes a general framework for leveraging unstructured pretraining in graph domains.** FIMP bridges the gap between foundation models and graph-based tasks by introducing a general-purpose framework for leveraging pretrained representations in graph learning. To our knowledge, no other framework currently provides the flexibility to utilize pretraining from arbitrary foundation models in graph-based tasks.
> 3. **Zero-shot Capabilities:** FIMP uniquely enables zero-shot inference on graph-structured data even when the pretrained models lack native graph support, as demonstrated in our experiments on the Mapillary image dataset. This capability allows FIMP to extend the utility of pretrained foundation models to graph domains without additional training—something not achievable with traditional GNNs or existing foundation model-based methods in non-textual graph settings.
>
> We hope this response clarifies the motivation and significance of our work. We encourage you to review the additional context and experiments provided in our responses to other reviewers, and we kindly ask you to reconsider your evaluation. Thank you for your time and feedback.
>
> [1] Graph Inductive Biases in Transformers without Message Passing. Ma et. al. 2023

---

> ### Author Response · Authors · 2024-12-02
>
> Thank you again for your initial feedback on our paper. During the rebuttal process, we have worked to clarify the contributions of FIMP and have substantially improved our experimental results by including additional baselines such as GRIT and GSN in our response to reviewer **uKm7**. These updates address concerns about the novelty, motivation, and significance of our work.
>
> Since this is the final day for reviewer comments, we would greatly appreciate your additional thoughts on our clarifications and revisions. Your engagement and feedback would be valuable in helping us further improve the manuscript.

---

### Official Review · Reviewer_ZuVo · 2024-11-08

**Soundness:** 3
**Presentation:** 3
**Contribution:** 4
**Rating:** 6
**Confidence:** 4

**Summary:**

The authors introduce Foundation-Informed Message Passing (FIMP), a novel message-passing framework designed to leverage pretrained non-textual foundation models for graph neural networks (GNNs).
Unlike existing GNNs, which utilize a single embedding for nodes, FIMP represents nodes as sequences of feature tokens.
This enables cross-node attention-based message-passing using the self-attention layers from pretrained models.
The proposed method is evaluated on tasks across image networks, spatial transcriptomics, and fMRI brain activity recordings, showing significant improvements over baseline GNNs.
Additionally, FIMP demonstrates zero-shot capabilities on image classification tasks using pretrained ViT.

**Strengths:**

I think it seems like the authors are trying hard to emphasize how FIMP is fundamentally different from existing attention-based GNNs like GATs and graph transformers. The statement with boldface might indicate that they have faced challenges in communicating the novelty of their work or distinguishing it from prior approaches in earlier submissions.
**However, in my opinion, this research is both timely and important because it aligns well with the current era of foundation models dominating various domains like NLP, computer vision, and even biomedicine.**
Existing foundation models excel with unstructured data (like text and images), but applying them directly to graph-structured data is challenging.
FIMP addresses this gap by allowing foundation models pretrained on unstructured data to be repurposed for graph-based tasks through its innovative tokenization and message-passing approach.
The authors provide comprehensive evaluations across diverse datasets, demonstrating that FIMP outperforms existing GNN models like GCN, GraphSAGE, and GATs. Notably, FIMP with pretrained models (e.g., scGPT and BrainLM) achieves the best results in gene expression prediction and brain activity reconstruction.
The zero-shot embedding experiments on the Mapillary image dataset highlight FIMP's ability to leverage pretrained foundation models without additional training, which is a significant advantage in scenarios where labeled graph data is scarce.

**Weaknesses:**

- While I acknowledge the novelty and potential impact of the research direction taken in this work, I must also express some concerns. The proposed module, despite its creative approach (direction), appears somewhat simplistic in its current form. It would benefit greatly from a deeper theoretical analysis to clarify why the architecture is effective and how it fundamentally differs from existing methods. This would not only strengthen the authors’ claims but also provide a clearer understanding of the model’s underlying principles.



- I noticed that while the paper benchmarks against various GNNs, it does not include comparisons with several main baselines (non-foundation models) that are well-established within each specific domain. Is there a particular reason why these comparisons were omitted? Including these baselines would provide a more comprehensive evaluation and help validate the effectiveness of FIMP beyond the GNN-specific context.


- I would like to address the reporting of experimental results. The performance metrics are averaged over five runs, but it is unclear if this number of repetitions is statistically sufficient to ensure robust conclusions, especially given the variability in results. For example, in the Mouse Embryo cell type classification task, the reported standard deviations appear quite significant. Increasing the number of runs or providing a justification for why five is adequate would enhance the reliability of the reported outcomes.


- (minor) Additionally, regarding the reference to Graph Transformer Networks (GTNs) on line 803, I believe there might be a mischaracterization of the original work by Yun et al. (2019). The GTNs discussed in that paper are not graph transformers in the sense typically understood in this context. Instead, they are better described as a graph-adapted version of Spatial Transformer Networks [1]. As stated by Yun et al.:
> Yun et al. (2019): "GTNs can be viewed as a graph analogue of Spatial Transformer Networks which explicitly learn spatial transformations of input images or features."

[1] M. Jaderberg, K. Simonyan, A. Zisserman, et al. Spatial transformer networks. In Advances in neural information processing systems, pages 2017–2025, 2015.

**Questions:**

please see the above section.

---

> ### Author Response · Authors · 2024-11-19
> **Rebuttal, Part 1**
>
> We appreciate your thoughtful and enthusiastic review of our submission, and we are glad that you recognize the novelty and importance of FIMP in leveraging pretrained non-textual foundation models for graph-based tasks. Below, we address some of the concerns and suggestions you raised:
>
> ### **Theoretical Analysis of FIMP’s Effectiveness**
> Thank you for your suggestion to include a deeper theoretical analysis of why the architecture is effective! We would like to provide additional insights into the theoretical underpinnings of FIMP:
> 1. **Token Representation:** Foundation models, particularly those based on transformers, have demonstrated a strong capacity to learn hierarchical and relational patterns in data. Tokenizing nodes into sequences of feature vectors allows FIMP to capture rich structural and relational features at a finer granularity than traditional Graph Neural Networks (GNNs). This approach is analogous to how transformers capture dependencies in sequences of words or image patches.
>     - It is well established in fields such as Natural Language Processing (NLP) that tokenization and pretraining on large corpora of data provides large benefits in terms of performance and generalizability. De Santis et al. (2024) directly compared transformer-based models employing word embedding against traditional bag-of-word approaches for user categorization based on online posts, demonstrating that transformer-based models utilizing word embedding outperform traditional bag-of-words approaches. The key insight is that tokenizing inputs allows transformers to preserve the original word ordering and contextualize different parts of input sentences, better capturing the meaning of input text compared to traditional bag-of-words approaches which lose positional information. This is true in Computer Vision as well, with Vision Transformers [2] demonstrating that patch tokenization can allow transformers to be pretrained effectively at large scale to achieve state-of-the-art performance on Computer Vision tasks.
>     - We view FIMP’s tokenization with the same benefits in mind, since FIMP allows for feature-level tokenization and contextualization of node entities in graphs. This is not possible with traditional GNNs, which operate over node feature vectors much like traditional bag-of-word models operated in NLP before transformers and word tokenization became widespread. This bag-of-words approach can be seen in common GNN benchmark datasets such as Cora [3], OGB [4], and Amazon [5].
>
> 2. **Transferability and Generalization:** The pretrained representations in foundation models, learned from vast datasets, generalize well across downstream tasks. Encoding nodes as token sequences enables FIMP to utilize these pretrained representations effectively, improving performance on downstream graph-based tasks.
>
> 3. **Feature-Level Interactions:** FIMP’s cross-node attention operates at the level of feature sequences rather than single node embeddings. This allows for detailed feature-level interactions between neighboring nodes, which traditional GNNs cannot achieve, and provides more expressive interaction between neighboring nodes during message passing.
>
> We will incorporate these theoretical motivations into the revised manuscript to strengthen the understanding of FIMP's principles and its distinctions from prior works.
>
> **References**
> [1] De Santis, Enrico, et al. "From Bag-of-Words to Transformers: A Comparative Study for Text Classification in Healthcare Discussions in Social Media." IEEE Transactions on Emerging Topics in Computational Intelligence (2024).
> [2] Dosovitskiy, Alexey. "An image is worth 16x16 words: Transformers for image recognition at scale." arXiv preprint arXiv:2010.11929 (2020).
> [3] Yang, Zhilin, William Cohen, and Ruslan Salakhudinov. "Revisiting semi-supervised learning with graph embeddings." International conference on machine learning. PMLR, 2016.
> [4] Hu, Weihua, et al. "Open graph benchmark: Datasets for machine learning on graphs." Advances in neural information processing systems 33 (2020): 22118-22133.
> [5] Shchur, Oleksandr, et al. "Pitfalls of graph neural network evaluation." arXiv preprint arXiv:1811.05868 (2018).

---

> > ### Author Response · Authors · 2024-11-19
> > **Rebuttal, Part 2**
> >
> > ### **Comparisons with Domain-Specific Baselines**
> > We appreciate your concern regarding the inclusion of more domain-specific baselines. In the current version, we compare FIMP against both GNN baselines as well as the pretrained foundation models themselves (scGPT, ViT, and BrainLM) with the goal of highlighting that the application of these foundation models in graph-based settings outperforms traditional GNN models as well as the foundation model itself on the graph-based task.
> >
> > We acknowledge the value of adding other domain-specific baselines, however we believe that the foundation models themselves represent strong baselines in their respective domains. In future work, we aim to incorporate domain-specific baselines for in-depth evaluations in particular data domains.
> >
> > ### **Statistical Robustness of Experimental Results**
> > We completely agree that sufficient experiment repetitions are crucial for ensuring statistical robustness. We chose to average results over five runs due to compute limitations, however we note that for most tasks, the standard deviations are smaller than the performance gap between FIMP and other GNN baselines, giving FIMP a healthy performance margin in many metrics.
> >
> > On tasks with higher variability across runs, such as the Mouse Embryo classification task, we attribute the higher variability to dataset-specific factors such as class imbalance. To address this, we are running additional repetitions on this task, which we hope to complete this week. We believe this will strengthen the reliability of our reported outcomes.
> >
> > ### **Mischaracterization of Graph Transformer Networks (GTNs)**
> > Thank you for catching this! We agree that the GTNs proposed in Yun et al. (2019) are more closely related to Spatial Transformer Networks than to the typical graph transformers discussed in the GNN literature. We will revise the manuscript to accurately reflect this distinction and add additional references of traditional Graph Transformer networks in the related works to avoid any potential mischaracterizations.
> >
> >
> > We hope these responses address your concerns and further clarify the contributions and significance of FIMP. We deeply appreciate your positive evaluation and your acknowledgment of the novelty, timeliness, and potential impact of our work. We look forward to continuing the discussion and welcome any additional feedback you may have.

---

> > > ### Comment · Reviewer_ZuVo · 2024-11-26
> > >
> > > Thank you for your detailed rebuttal and thoughtful responses to my concerns.
> > > I appreciate the additional insights into the theoretical foundations of FIMP, particularly the parallels drawn with tokenization in NLP and Vision Transformers.
> > > These clarifications strengthen the motivation behind the design choices and highlight the model’s potential to leverage pretrained foundation models for graph-based tasks effectively.
> > > I also acknowledge your explanation regarding the omission of domain-specific baselines and the compute limitations for experiment repetitions.
> > > While these factors are understandable, incorporating these elements in future work will be essential for demonstrating the broader applicability and robustness of FIMP.
> > > Finally, I appreciate the correction regarding the characterization of GTNs and your commitment to revising the manuscript.
> > > After carefully considering your rebuttal and other reviews, I maintain my initial score as I believe FIMP offers an interesting approach to leveraging foundation models for graph-based tasks, albeit with room for further refinement in some areas.

---

> ### Author Response · Authors · 2024-11-26
> **Mouse Embryo Experiment Repeats**
>
> Thank you for your reply! We are pleased to share the results of additional experiment runs for the mouse embryo classification task, which aimed to improve the statistical robustness of this particular task. We increased the number of experiment repetitions from 5 to 10 for most models by adding runs with new random seeds. These updated results are summarized below:
>
> | Method | Accuracy | F1-score |
> | ----- | ----- | ----- |
> | GCN | 75.78 ± 2.462 | 0.738 ± 0.026 |
> | GraphSAGE | 80.50 ± 2.701 | 0.795 ± 0.028 |
> | GAT | 74.33 ± 2.824 | 0.726 ± 0.027 |
> | GIN | 74.06 ± 2.304 | 0.726 ± 0.025 |
> | GPS | 82.62 ± 2.469 | 0.819 ± 0.026 |
> | FIMP-base | 81.33 ± 2.633 | 0.807 ± 0.274 |
> | scGPT | 82.93 ± 0.419 | 0.820 ± 0.005 |
> | FIMP-scGPT | **83.66 ± 2.631** | **0.824 ± 0.025** |
>
> The rebuttal time constraints prevented us from completing all 5 new repetitions for FIMP-scGPT and scGPT, however the additional runs we were able to add suggest that the relative performance relationships between FIMP and the baseline models remain consistent. This indicates that our reported results are robust.
>
> We appreciate your engagement throughout the review process! Please let us know if there are remaining points we have failed to address.

---

### Meta-Review · Area_Chair_LVXW · 2024-12-10

**Metareview:**

This paper introduces a novel message-passing framework by incorporating existing non-textural pre-trained models for graph-structured tasks. To this end, it employs multi-channel cross-attention to create the message. The performance is verified on multiple fields. Although the proposed model is clear and universal to different fields, its novelty and motivation are of concern. Firstly, the pre-trained models are only used as the initial node feature without any improvements. Secondly, the cross-attention is not novel, although the multi-channel variant is introduced. Thirdly, the graph structure, which is the most important of graph learning, is not properly considered. Therefore, it is not good enough for published.

**Additional Comments On Reviewer Discussion:**

Three of the four reviewers engaged in the rebuttal. Although some of them raise the score, two of them believe it is marginally below the acceptance threshold. Besides, I also believe reviewer YLzx, who did not engage in the rebuttal, points out some serious issues in this paper. Thus, I tend to reject this paper.

---

### Decision · Program_Chairs · 2025-01-22

Reject